# Approximate Cross-Validation with Low-Rank Data in High Dimensions

**William T. Stephenson**
MIT
wtstephe@mit.edu

**Madeleine Udell**
Cornell University
udell@cornell.edu

**Tamara Broderick**
MIT
tamarab@mit.edu

## Abstract

Many recent advances in machine learning are driven by a challenging trifecta: large data size $N$, high dimensions, and expensive algorithms. In this setting, cross-validation (CV) serves as an important tool for model assessment. Recent advances in approximate cross validation (ACV) provide accurate approximations to CV with only a single model fit, avoiding traditional CV's requirement for repeated runs of expensive algorithms. Unfortunately, these ACV methods can lose both speed and accuracy in high dimensions — unless sparsity structure is present in the data. Fortunately, there is an alternative type of simplifying structure that is present in most data: approximate low rank (ALR). Guided by this observation, we develop a new algorithm for ACV that is fast and accurate in the presence of ALR data. Our first key insight is that the Hessian matrix — whose inverse forms the computational bottleneck of existing ACV methods — is ALR. We show that, despite our use of the *inverse* Hessian, a low-rank approximation using the largest (rather than the smallest) matrix eigenvalues enables fast, reliable ACV. Our second key insight is that, in the presence of ALR data, error in existing ACV methods roughly grows with the (approximate, low) rank rather than with the (full, high) dimension. These insights allow us to prove theoretical guarantees on the quality of our proposed algorithm — along with fast-to-compute upper bounds on its error. We demonstrate the speed and accuracy of our method, as well as the usefulness of our bounds, on a range of real and simulated data sets.

## 1   Introduction

Recent machine learning advances are driven at least in part by increasingly rich data sets — large in both data size $N$ and dimension $D$. The proliferation of data and algorithms makes cross-validation (CV) [Stone, 1974, Geisser, 1975, Musgrave et al., 2020] an appealing tool for model assessment due its ease of use and wide applicability. For high-dimensional data sets, leave-one-out CV (LOOCV) is often especially accurate as its folds more closely match the true size of the data [Burman, 1989]; see also Figure 1 of Rad and Maleki [2020]. Traditionally many practitioners nonetheless avoid LOOCV due its computational expense; it requires re-running an expensive machine learning algorithm $N$ times. To address this expense, a number of authors have proposed approximate cross-validation (ACV) methods [Beirami et al., 2017, Rad and Maleki, 2020, Giordano et al., 2019]; these methods are fast to run on large data sets, and both theory and experiments demonstrate their accuracy. But these methods struggle in high-dimensional problems in two ways. First, they require inversion of a $D \times D$ matrix, a computationally expensive undertaking. Second, their accuracy can degrade in high dimensions; see Fig. 1 of Stephenson and Broderick [2020] for a classification example and Fig. 1 below for a count-valued regression example. Koh and Liang [2017], Lorraine et al. [2020] have investigated approximations to the matrix inverse for problems similar to ACV, but these approximations do not work well for ACV itself; see [Stephenson and Broderick, 2020, Appendix B].

Stephenson and Broderick [2020] demonstrate how a practitioner might avoid these high-dimensional problems in the presence of sparse data. But sparsity may be a somewhat limiting assumption.

We here consider approximately *low-rank* (ALR) data. Udell and Townsend [2019] argue that ALR data matrices are pervasive in applications ranging from fluid dynamics and genomics to social networks and medical records — and that there are theoretical reasons to expect ALR structure in many large data matrices. For concreteness and to facilitate theory, we focus on fitting generalized linear models (GLMs). We note that GLMs are a workhorse of practical data analysis; as just one example, one of many popular books on GLMs [McCullagh, 1989] has been cited over 9,000 times since 2015 as of this writing. While accurate ACV methods for GLMs alone thus have potential for great impact, we expect many of our insights may extend beyond both GLMs and LOOCV (i.e. to other CV and bootstrap-like "retraining" schemes).

In particular, we propose an algorithm for fast, accurate ACV for GLMs with high-dimensional covariate matrices — and provide computable upper bounds on the error of our method relative to exact LOOCV. Two major innovations power our algorithm. First, we prove that existing ACV methods automatically obtain high accuracy in the presence of high-dimensional yet ALR data. Our theory provides cheaply computable upper bounds on the error of existing ACV methods. Second, we notice that the $D \times D$ matrix that needs to be inverted in ACV is ALR when the covariates are ALR. We propose to use a low-rank approximation to this matrix. We provide a computable upper bound on the extra error introduced by using such a low-rank approximation. By studying our bound, we show the surprising fact that, for the purposes of ACV, the matrix is well approximated by using its *largest* eigenvalues, despite the fact that ACV uses the matrix inverse. We demonstrate the speed and accuracy of both our method and bounds with a range of experiments.

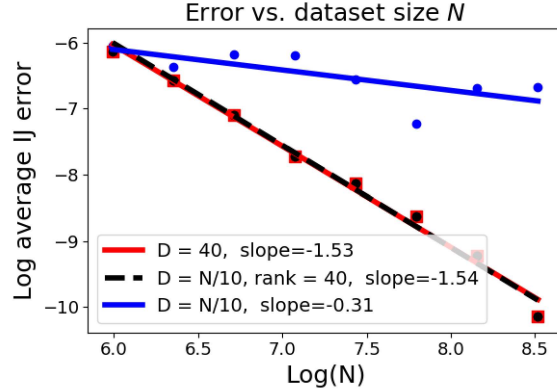

Figure 1: Accuracy of the IJ approximation in Eq. (4) for a synthetic Poisson regression problem versus the dataset size $N$. Red shows the accuracy when the data dimension is fixed at $D = 40$, blue when the dimension grows as $D = N/10$, and black when the dimension grows as $D = N/10$ but with a fixed rank of 40. High-dimensional yet low-rank data has identical performance to low-dimensional data.

## 2 Background: approximate CV methods

We consider fitting a generalized linear model (GLM) with parameter $\theta \in \mathbb{R}^D$ to some dataset with $N$ observations, $\{x_n, y_n\}_{n=1}^N$, where $x_n \in \mathbb{R}^D$ are covariates and $y_n \in \mathbb{R}$ are responses. We suppose that the $x_n$ are approximately low rank (ALR); that is, the matrix $X \in \mathbb{R}^{N \times D}$ with rows $x_n$ has many singular values near zero. These small singular values can amplify noise in the responses. Hence it is common to use $\ell_2$ regularization to ensure that our estimated parameter $\hat{\theta}$ is not too sensitive to the subspace with small singular values; the rotational invariance of the $\ell_2$ regularizer automatically penalizes any deviation of $\theta$ away from the low-rank subspace [Hastie et al., 2009, Sec. 3.4]. Thus we consider:

$$\hat{\theta} := \underset{\theta \in \mathbb{R}^D}{\arg\min} \frac{1}{N} \sum_{n=1}^N f(x_n^T \theta, y_n) + \frac{\lambda}{2} \|\theta\|_2^2, \tag{1}$$

where $\lambda \geq 0$ is some regularization parameter and $f : \mathbb{R} \times \mathbb{R} \to \mathbb{R}$ is convex in its first argument for each $y_n$. Throughout, we assume $f$ to be twice differentiable in its first argument. To use leave-one-out CV (LOOCV), we compute $\hat{\theta}_{\backslash n}$, the estimate of $\theta$ after deleting the $n$th datapoint from the sum, for each $n$. To assess the out-of-sample error of our fitted $\hat{\theta}$, we then compute:

$$\frac{1}{N} \sum_{n=1}^N \text{Err}(x_n^T \hat{\theta}_{\backslash n}, y_n), \tag{2}$$

where $\text{Err} : \mathbb{R} \times \mathbb{R} \to \mathbb{R}$ is some function measuring the discrepancy between the observed $y_n$ and its prediction based on $\hat{\theta}_{\backslash n}$ — for example, squared error or logistic loss.

Computing $x_n^T \hat{\theta}_{\backslash n}$ for every $n$ requires solving $N$ optimization problems, which can be a prohibitive computational expense. Approximate CV (ACV) methods aim to alleviate this burden via one of two principal approaches below. Denote the Hessian of the objective by $H := (1/N) \sum_{n=1}^N \nabla_\theta^2 f(x_n^T \hat{\theta}, y_n) + \lambda I_D$ and the $k$th scalar derivative of $f$ as $\hat{D}_n^{(k)} := d^k f(z, y_n)/dz^k \big|_{z=x_n^T \hat{\theta}}$. Finally, let $Q_n := x_n^T H^{-1} x_n$ be the $n$th quadratic form on $H^{-1}$. The first approximation, based on taking a Newton step from $\hat{\theta}$ on the objective $(1/N) \sum_{m \neq n}^N f(x_m^T \theta, y_m) + \lambda \|\theta\|_2^2$, was proposed by Obuchi and Kabashima [2016, 2018], Rad and Maleki [2020], Beirami et al. [2017]. We denote this approximation by $\text{NS}_{\backslash n}$; specializing to GLMs, we have:

$$x_n^T \hat{\theta}_{\backslash n} \approx x_n^T \text{NS}_{\backslash n} := x_n^T \hat{\theta} + \frac{\hat{D}_n^{(1)}}{N} \frac{Q_n}{1 - \hat{D}_n^{(2)} Q_n}. \tag{3}$$

Throughout we focus on approximating $x_n^T \hat{\theta}_{\backslash n}$, rather than $\hat{\theta}_{\backslash n}$, since $x_n^T \hat{\theta}_{\backslash n}$ is the argument of Eq. (2). See Appendix A for a derivation of Eq. (3). The second approximation we consider is based on the infinitesimal jackknife [Jaeckel, 1972, Efron, 1982]; it was conjectured as a possible ACV method by Koh and Liang [2017], used in a comparison by Beirami et al. [2017], and studied in depth by Giordano et al. [2019]. We denote this approximation by $\text{IJ}_{\backslash n}$; specializing to GLMs, we have:

$$x_n^T \hat{\theta}_{\backslash n} \approx x_n^T \text{IJ}_{\backslash n} := x_n^T \hat{\theta} + (\hat{D}_n^{(1)}/N) Q_n. \tag{4}$$

See Appendix A for a derivation. We consider both $\text{NS}_{\backslash n}$ and $\text{IJ}_{\backslash n}$ in what follows as the two have complementary strengths. In our experiments in Section 6, $\text{NS}_{\backslash n}$ tends to be more accurate; we suspect that GLM users should generally use $\text{NS}_{\backslash n}$. On the other hand, $\text{NS}_{\backslash n}$ requires the inversion of a different $D \times D$ matrix for each $n$. In the case of LOOCV for GLMs, each matrix differs by a rank-one update, so standard matrix inverse update formulas allow us to derive Eq. (3), which requires only a single inverse across folds. But such a simplification need not generally hold for models beyond GLMs and data re-weightings beyond LOOCV (such as other forms of CV or the bootstrap). By contrast, even beyond GLMs and LOOCV, the IJ requires only a single matrix inverse for all $n$.

In any case, we notice that existing theory and experiments for both $\text{NS}_{\backslash n}$ and $\text{IJ}_{\backslash n}$ tend to either focus on low dimensions or show poor performance in high dimensions; see Appendix C for a review. One problem is that error in both approximations can grow large in high dimensions. See [Stephenson and Broderick, 2020] for an example; also, in Fig. 1, we show the $\text{IJ}_{\backslash n}$ on a synthetic Poisson regression task. When we fix $D = 40$ and $N$ grows, the error drops quickly; however, if we fix $D/N = 1/10$ the error is substantially worse. A second problem is that both $\text{NS}_{\backslash n}$ and $\text{IJ}_{\backslash n}$ rely on the computation of $Q_n = x_n^T H^{-1} x_n$, which in turn relies on computation[1] of $H^{-1}$. The resulting $O(D^3)$ computation time quickly becomes impractical in high dimensions. Our major contribution is to show that both of these issues can be avoided when the data are ALR.

## 3  Methodology

We now present our algorithm for fast, approximate LOOCV in GLMs with ALR data. We then state our main theorem, which (1) bounds the error in our algorithm relative to exact CV, (2) gives the computation time of our algorithm, and (3) gives the computation time of our bounds. Finally we discuss the implications of our theorem before moving on to the details of its proof in the remainder of the paper.

Our method appears in Algorithm 1. To avoid the $O(D^3)$ matrix inversion cost, we replace $H$ by $\widetilde{H} \approx H$, where $\widetilde{H}$ uses a rank-$K$ approximation and can be quickly inverted. We can then use $\widetilde{H}$ to compute $\widetilde{Q}_n \approx Q_n$, which enters into either the NS or IJ approximation, as desired.

Before stating Theorem 1, we establish some notation. We will see in Proposition 2 of Section 4 that we can provide computable upper bounds $\eta_n \geq |\widetilde{Q}_n - Q_n|$; $\eta_n$ will enter directly into the error

**Algorithm 1** Approximation to $\{x_n^T \hat{\theta}_{\setminus n}\}_{n=1}^N$ for low-rank GLMs
---
1: **procedure** APPXLOOCV($\hat{\theta}, X, \lambda, \{\hat{D}_n^{(1)}\}_{n=1}^N, \{\hat{D}_n^{(2)}\}_{n=1}^N, K$)
2:     $B \leftarrow X^T \text{diag}\{\hat{D}_n^{(2)}\}_{n=1}^N X$                    ▷ The Hessian, $H$, equals $B + \lambda I_D$
3:     $\{\tilde{Q}_n\}_{n=1}^N \leftarrow$ APPXQN($B, K, \lambda$)        ▷ Uses rank-$K$ decomposition of $B$ (Section 5)
4:     **for** $n = 1, \ldots, N$ **do**
5:         **either** $x_n^T \widetilde{\text{NS}}_{\setminus n} \leftarrow x_n^T \text{NS}_{\setminus n}(\tilde{Q}_n)$        ▷ i.e., compute Eq. (3) using $\tilde{Q}_n$ instead of $Q_n$
6:         **or** $x_n^T \widetilde{\text{IJ}}_{\setminus n} \leftarrow x_n^T \text{IJ}_{\setminus n}(\tilde{Q}_n)$        ▷ i.e., compute Eq. (4) using $\tilde{Q}_n$ instead of $Q_n$
7:     **end for**
8:     **return** $\{x_n^T \widetilde{\text{NS}}_{\setminus n}\}_{n=1}^N$ **or** $\{x_n^T \widetilde{\text{IJ}}_{\setminus n}\}_{n=1}^N$                    ▷ User's choice
9: **end procedure**
---

bound for $x_n^T \widetilde{\text{IJ}}_{\setminus n}$ in Theorem 1 below. To bound the error of $x_n^T \widetilde{\text{NS}}_{\setminus n}$, we need to further define

$$E_n := \max\left\{ \left| \frac{\tilde{Q}_n + \eta_n}{1 - \hat{D}_n^{(2)}(\tilde{Q}_n + \eta_n)} - \frac{\tilde{Q}_n}{1 - \hat{D}_n^{(2)}\tilde{Q}_n} \right|, \left| \frac{\tilde{Q}_n - \eta_n}{1 - \hat{D}_n^{(2)}(\tilde{Q}_n - \eta_n)} - \frac{\tilde{Q}_n}{1 - \hat{D}_n^{(2)}\tilde{Q}_n} \right| \right\}.$$

Additionally, we will see in Proposition 1 of Section 4 that we can bound the "local Lipschitz-ness" of the Hessian related to the third derivatives of $f$ evaluated at some $z$, $\hat{D}_n^{(3)}(z) := d^3 f(z, y_n)/dz^3|_{z=z}$. We will denote our bound by $M_n$:

$$M_n \geq \left( \frac{1}{N} \sum_{m \neq n} \|x_m\|_2^2 \right) \max_{s \in [0,1]} \left| \hat{D}_n^{(3)} \left( x_n^T((1-s)\hat{\theta} + s\hat{\theta}_{\setminus n}) \right) \right|, \tag{5}$$

We are now ready to state, and then discuss, our main result — which is proved in Appendix D.3.

**Theorem 1.** *(1) Accuracy: Let $\eta_n \geq |Q_n - \tilde{Q}_n|$ be the upper bound produced by Proposition 2 and $M_n$ the local Lipschitz constants computed in Proposition 1. Then the estimates $x_n^T \widetilde{\text{NS}}_{\setminus n}$ and $x_n^T \widetilde{\text{IJ}}_{\setminus n}$ produced by Algorithm 1 satisfy:*

$$|x_n^T \widetilde{\text{NS}}_{\setminus n} - x_n^T \hat{\theta}_{\setminus n}| \leq \frac{M_n}{N^2 \lambda^3} |\hat{D}_n^{(1)}|^2 \|x_n\|_2^3 + |\hat{D}_n^{(1)}| E_n \tag{6}$$

$$|x_n^T \widetilde{\text{IJ}}_{\setminus n} - x_n^T \hat{\theta}_{\setminus n}| \leq \frac{M_n}{N^2 \lambda^3} |\hat{D}_n^{(1)}|^2 \|x_n\|_2^3 + \frac{1}{N^2 \lambda^2} |\hat{D}_n^{(1)}||\hat{D}_n^{(2)}| \|x_n\|_2^4 + |\hat{D}_n^{(1)}| \eta_n. \tag{7}$$

*(2) Algorithm computation time: The runtime of Algorithm 1 is in $O(NDK + K^3)$. (3) Bound computation time: The upper bounds in Eqs. (6) and (7) are computable in $O(DK)$ time for each $n$ for common GLMs such as logistic and Poisson regression.*

To interpret the running times, note that standard ACV methods have total runtime in $O(ND^2 + D^3)$. So Algorithm 1 represents a substantial speedup when the dimension $D$ is large and $K \ll D$. Also, note that our bound computation time has no worse behavior than our algorithm runtime. We demonstrate in our experiments (Section 6) that our error bounds are both computable and useful in practice. To help interpret the bounds, note that they contain two sources of error: (A) the error of our additional approximation relative to existing ACV methods (i.e. the use of $\tilde{Q}_n$) and (B) the error of existing ACV methods in the presence of ALR data. Our first corollary notes that (A) goes to zero as the data becomes exactly low rank.

**Corollary 1.** *As the data becomes exactly low rank with rank $R$ (i.e., $X$'s lowest singular values $\sigma_d \to 0$ for $d = R+1, \ldots, D$), we have $\eta_n, E_n \to 0$ if $K \geq R$.*

See Appendix D.4 for a proof. Our second corollary gives an example for which the error in existing (exact) ACV methods (B) vanishes as $N$ grows.

**Corollary 2.** *Suppose the third derivatives $\hat{D}_n^{(3)}$ and the $x_n$ are both bounded and the data are exactly low-rank with constant rank $R$. Then with $N \to \infty$, $D$ growing at any rate, and $K$ arbitrary, the right hand sides of Eqs. (6) and (7) reduce to $|\hat{D}_n^{(1)}| E_n$ and $|\hat{D}_n^{(1)}| \eta_n$, respectively.*

We note that Corollary 2 is purely illustrative, and we strongly suspect that none of its conditions are necessary. Indeed, our experiments in Section 6 show that the bounds of Theorem 1 imply reasonably low error for non-bounded derivatives with ALR data and only moderate $N$.

# 4 Accuracy of exact ACV with approximately low-rank data

Recall that the main idea behind Algorithm 1 is to compute a fast approximation to existing ACV methods by exploiting ALR structure. To prove our error bounds, we begin by proving that the exact ACV methods $\mathrm{NS}_{\setminus n}$ and $\mathrm{IJ}_{\setminus n}$ approximately (respectively, exactly) retain the low-dimensional accuracy displayed in red in Fig. 1 when applied to GLMs with approximately (respectively, exactly) low-rank data. Let us first define low-rank data. Let $X = U\Sigma V^T$ be the singular value decomposition of $X$, where $U \in \mathbb{R}^{N \times D}$ has orthonormal columns, $\Sigma \in \mathbb{R}^{D \times D}$ is a diagonal matrix, and $V \in \mathbb{R}^{D \times D}$ is an orthonormal matrix.

**Definition 1.** *We say that a matrix $X$ with singular value decomposition $X = U\Sigma V$ is of* exactly *low-rank $R$ if $\Sigma_{dd} = 0$ for all $d > R$. We say that $X$ is of* approximately low rank (ALR) $R$ if $\Sigma_{dd} \approx 0$ for all $d > R$.

We note that this definition of ALR is different from that in Udell and Townsend [2019], which we gave in Section 1 as a motivation for considering ALR data. In particular, Udell and Townsend [2019] define a matrix $X$ to be of ALR if it is entry-wise $\varepsilon$-close to some matrix of exactly low-rank $R$; such a matrix can be very different from our definition of ALR, as such a matrix can have $\Sigma_{R+1,R+1} = D\varepsilon$. While we only consider Udell and Townsend [2019] as general motivation, we note that in our work below, we will consider the case of $\Sigma_{dd} \to 0$, making the two definitions of ALR equivalent.

We now show that existing ACV methods are accurate in the presence of exactly low-rank data. Let $V_{:R}$ be the top $R$ right singular vectors of $X$ (i.e. the first $R$ columns of $V$), and fit a model restricted to $R$ dimensions as:

$$\hat{\phi} := \arg\min_{\phi \in \mathbb{R}^R} \frac{1}{N} \sum_{n=1}^N f((V_{:R}^T x_n)^T \phi) + \frac{\lambda}{2}\|\phi\|_2^2.$$

Let $\hat{\phi}_{\setminus n}$ be the $n$th leave-one-out parameter estimate from this problem, and let $\mathrm{RNS}_{\setminus n}$ and $\mathrm{RIJ}_{\setminus n}$ be Eq. (3) and Eq. (4) applied to this restricted problem. We can now show that the error of $\mathrm{IJ}_{\setminus n}$ and $\mathrm{NS}_{\setminus n}$ applied to the full $D$-dimensional problem is exactly the same as the error of $\mathrm{RNS}_{\setminus n}$ and $\mathrm{RIJ}_{\setminus n}$ applied to the restricted $R \ll D$ dimensional problem.

**Lemma 1.** *Assume that the data matrix $X$ is exactly low-rank $R$. Then $|x_n^T \mathrm{NS}_{\setminus n} - x_n^T \hat{\theta}_{\setminus n}| = |(V_{:R}^T x_n)^T \mathrm{RNS}_{\setminus n} - (V_{:R}^T x_n)^T \hat{\phi}_{\setminus n}|$ and $|x_n^T \mathrm{IJ}_{\setminus n} - x_n^T \hat{\theta}_{\setminus n}| = |(V_{:R}^T x_n)^T \mathrm{RIJ}_{\setminus n} - (V_{:R}^T x_n)^T \hat{\phi}_{\setminus n}|$.*

See Appendix D.1 for a proof. Based on previous work (e.g., [Beirami et al., 2017, Rad and Maleki, 2020, Giordano et al., 2019]), we expect the ACV errors $|(V_{:R}^T x_n)^T \mathrm{RNS}_{\setminus n} - (V_{:R}^T x_n)^T \hat{\phi}_{\setminus n}|$ and $|(V_{:R}^T x_n)^T \mathrm{RIJ}_{\setminus n} - (V_{:R}^T x_n)^T \hat{\phi}_{\setminus n}|$ to be small, as they represent the errors of $\mathrm{NS}_{\setminus n}$ and $\mathrm{IJ}_{\setminus n}$ applied to an $R$-dimensional problem. We confirm Lemma 1 numerically in Fig. 1, where the error for the $D = 40$ problems (red) exactly matches that of the high-$D$ but exact low-rank $R = 40$ problems (black).

However, real-world covariate matrices $X$ are rarely exactly low-rank. By adapting results from Wilson et al. [2020], we can give bounds that smoothly decay as we leave the exact low-rank setting of Lemma 1. To that end, define:

$$L_n := \left( \frac{1}{N} \sum_{m:\, m \neq n}^N \|x_m\|_2^2 \right) \max_{s \in [0,1]} \hat{D}_n^{(3)} \left( x_n^T ((1-s)\hat{\theta} + s\hat{\theta}_{\setminus n}) \right). \tag{8}$$

**Lemma 2.** *Assume that $\lambda > 0$. Then, for all $n$:*

$$|x_n^T \mathrm{NS}_{\setminus n} - x_n^T \hat{\theta}_{\setminus n}| \leq \frac{L_n}{N^2 \lambda^3} |\hat{D}_n^{(1)}|^2 \|x_n\|_2^3 \tag{9}$$

$$|x_n^T \mathrm{IJ}_{\setminus n} - x_n^T \hat{\theta}_{\setminus n}| \leq \frac{L_n}{N^2 \lambda^3} |\hat{D}_n^{(1)}|^2 \|x_n\|_2^3 + \frac{1}{N^2 \lambda^2} |\hat{D}_n^{(1)}| |\hat{D}_n^{(2)}| \|x_n\|_2^4. \tag{10}$$

*Furthermore, these bounds continuously decay as the data move from exactly to approximately low rank in that they are continuous in the singular values of $X$.*

The proofs of Eqs. (9) and (10) mostly follow from results in Wilson et al. [2020], although our results removes a Lipschitz assumption on the $\hat{D}_n^{(2)}$; see Appendix D.2 for a proof.

Our bounds are straightforward to compute; we can calculate the norms $\|x_n\|_2$ and evaluate the derivatives $\hat{D}_n^{(1)}$ and $\hat{D}_n^{(2)}$ at the known $x_n^T \hat{\theta}$. The only unknown quantity is $L_n$. However, we can upper bound the $L_n$ using the following proposition.

**Proposition 1.** *Let $\mathcal{Z}_n$ be the set of $z \in \mathbb{R}$ such that $|z| \leq |x_n^T \hat{\theta}| + |\hat{D}_n^{(1)}| \|x_n\|_2^2 / (N\lambda)$. For $L_n$ as defined in Eq. (8), we have the upper bound:*

$$L_n \leq M_n := \max_{z \in \mathcal{Z}_n} |\hat{D}_n^{(3)}(z)| \left( \frac{1}{N} \sum_{m:\, m \neq n}^{N} \|x_m\|_2^2 \right). \tag{11}$$

To compute an upper bound on the $M_n$ in turn, we can optimize $\hat{D}_n^{(3)}(z)$ for $|z| \leq |x_n^T \hat{\theta}| + |\hat{D}_n^{(1)}| \|x_n\|_2^2 / (N\lambda)$. This scalar problem is straightforward for common GLMs: for logistic regression, we can use the fact that $|\hat{D}_n^{(3)}| \leq 1/4$, and for Poisson regression with an exponential link function (i.e., $y_n \sim \text{Poisson}(\exp(x_n^T \theta)))$, we maximize $\hat{D}_n^{(3)}(z) = e^z$ with the largest $z \in \mathcal{Z}_n$.

## 5 Approximating the quadratic forms $Q_n$

---

**Algorithm 2** Estimate $Q_n = x_n^T (B + \lambda I_D)^{-1} x_n$ via a rank-$K$ decomposition of PSD matrix $B$. *Note*: as written, this procedure is not numerically stable. See Appendix E.3 for an equivalent but numerically stable version.

---
1: **procedure** APPXQN($B, K, \lambda$)
2:     **for** $k = 1, \ldots, K$ **do**
3:         $\mathcal{E}_k \leftarrow \mathcal{N}(0_D, I_D)$                        $\triangleright \; \mathcal{E} \in \mathbb{R}^{D \times K}$ has i.i.d. $\mathcal{N}(0,1)$ entries
4:     **end for**
5:     $\Omega \leftarrow$ ORTHONORMALIZECOLUMNS($\text{diag}\{1/(B_{dd} + \lambda)\}_{d=1}^{D} X^T X \mathcal{E}$)     $\triangleright$ Proposition 3
6:     $M \leftarrow B\Omega$
7:     $\widetilde{H} \leftarrow M(\Omega^T M)^{-1} M^T + \lambda I_D$              $\triangleright$ Rank-$K$ Nyström approximation of $B$
8:     **for** $n = 1, \ldots, N$ **do**
9:         $\widetilde{Q}_n \leftarrow \min\left\{ x_n^T \widetilde{H}^{-1} x_n, \; \|x_n\|_2^2 / (\lambda + \hat{D}_n^{(2)} \|x_n\|_2^2) \right\}$     $\triangleright$ Proposition 4
10:     **end for**
11:     **return** $\{\widetilde{Q}_n\}_{n=1}^N$
12: **end procedure**

---

The results of Section 4 imply that existing ACV methods achieve high accuracy on GLMs with ALR data. However, in high dimensions, the $O(D^3)$ cost of computing $H^{-1}$ in the $Q_n$ can be prohibitive. Koh and Liang [2017], Lorraine et al. [2020] have investigated an approximation to the matrix inverse for problems similar to ACV; however, in our experiments in Appendix B, we find that this method does not work well for ACV. Instead, we give approximations $\widetilde{Q}_n \approx Q_n$ in Algorithm 2 along with computable upper bounds on the error $|\widetilde{Q}_n - Q_n|$ in Proposition 4. When the data has ALR structure, so does the Hessian $H$; hence we propose a low-rank matrix approximation to $H$. This gives Algorithm 2 a runtime in $O(NDK + K^3)$, which can result in substantial savings relative to the $O(ND^2 + D^3)$ time required to exactly compute the $Q_n$. We will see that the main insights behind Algorithm 2 come from studying an upper bound on the approximation error when using a low-rank approximation.

Observe that by construction of $\Omega$ and $\widetilde{H}$ in Algorithm 2, the approximate Hessian $\widetilde{H}$ exactly agrees with $H$ on the subspace $\Omega$. We can compute an upper bound on the error $|x_n^T \widetilde{H}^{-1} x_n - Q_n|$ by recognizing that any error originates from components of $x_n$ orthogonal to $\Omega$:

**Proposition 2.** *Let $\lambda > 0$ and suppose there is some subspace $\mathcal{B}$ on which $H$ and $\widetilde{H}$ exactly agree: $\forall v \in \mathcal{B}, Hv = \widetilde{H}v$. Then $H^{-1}$ and $\widetilde{H}^{-1}$ agree exactly on the subspace $\mathcal{A} := H\mathcal{B}$, and*

$$|x_n^T \widetilde{H}^{-1} x_n - Q_n| \leq \frac{\left\| P_{\mathcal{A}}^{\perp} x_n \right\|_2^2}{\lambda}, \qquad \text{for all } n = 1, \ldots, N, \tag{12}$$

*where $P_{\mathcal{A}}^{\perp}$ denotes projection onto the orthogonal complement of $\mathcal{A}$.*

For a proof, see Appendix E.1. The bound from Eq. (12) is easy to compute in $O(DK)$ time given a basis for $\mathcal{B}$. It also motivates the choice of $\Omega$ in Algorithm 2. In particular, Proposition 3 shows that $\Omega$ approximates the rank-$K$ subspace $\mathcal{B}$ that minimizes the average of the bound in Eq. (12).

**Proposition 3.** *Let $V_{:K} \in \mathbb{R}^{D \times K}$ be the matrix with columns equal to the right singular vectors of $X$ corresponding to the $K$ largest singular values. Then the rank-$K$ subspace $\mathcal{B}$ minimizing $\sum_n \|P_\mathcal{A}^\perp x_n\|_2^2$ is an orthonormal basis for the columns of $H^{-1} V_{:K}$.*

*Proof.* $\sum_n \|P_\mathcal{A}^\perp x_n\|_2^2 = \|(I_D - P_\mathcal{A}) X^T\|_F^2$, where $\|\cdot\|_F$ denotes the Frobenius norm. Noting that the given choice of $\mathcal{B}$ implies that $\mathcal{A} = V_{:K}$, the result follows from the Eckart-Young theorem. $\square$

We now see that the choice of $\Omega$ in Algorithm 2 approximates the optimal choice $H^{-1} V_{:K}$. In particular, we use a single iteration of the subspace iteration method [Bathe and Wilson, 1973] to approximate $V_{:K}$ and then multiply by the diagonal approximation $\operatorname{diag} \{1/H_{dd}\}_{d=1}^D \approx H^{-1}$. This approximation uses the top singular vectors of $X$. We expect these directions to be roughly equivalent to the largest eigenvectors of $B := \sum_n \hat{D}_n^{(2)} x_n x_n^T$, which in turn are the largest eigenvectors of $H = B + \lambda I_D$. Thus we are roughly approximating $H$ by its *largest $K$* eigenvectors.

Why is it safe to neglect the small eigenvectors? At first glance this is strange, as to minimize the operator norm $\|H^{-1} - \widetilde{H}^{-1}\|_{op}$, one would opt to preserve the action of $H$ along its *smallest $K$* eigenvectors. The key intuition behind this reversal is that we are interested in the action of $H^{-1}$ in the direction of the datapoints $x_n$, which, on average, tend to lie near the largest eigenvectors of $H$.

Algorithm 2 uses one additional insight to improve the accuracy of its estimates. In particular, we notice that, by the definition of $H$, each $x_n$ lies in an eigenspace of $H$ with eigenvalue at least $\hat{D}_n^{(2)} \|x_n\|_2^2 + \lambda$. This observation undergirds the following result, which generates our final estimates $\widetilde{Q}_n \approx Q_n$, along with quickly computable bounds on their error $|\widetilde{Q}_n - Q_n|$.

**Proposition 4.** *The $Q_n = x_n^T H^{-1} x_n$ satisfy $0 < Q_n \leq \|x_n\|_2^2 / (\lambda + \hat{D}_n^{(2)} \|x_n\|_2^2)$. Furthermore, letting $\widetilde{Q}_n := \min\{x_n^T \widetilde{H}^{-1} x_n, \|x_n\|_2^2 / (\lambda + \hat{D}_n^{(2)} \|x_n\|_2^2)\}$, we have the error bound*

$$|\widetilde{Q}_n - Q_n| \leq \min \left\{ \frac{\|P_\mathcal{A}^\perp x_n\|_2^2}{\lambda}, \frac{\|x_n\|_2^2}{\lambda + \hat{D}_n^{(2)} \|x_n\|_2^2} \right\}. \tag{13}$$

See Appendix E.1 for a proof. We finally note that Algorithm 2 strongly resembles algorithms from the randomized numerical linear algebra literature. Indeed, the work of Tropp et al. [2017] was the original inspiration for Algorithm 2, and Algorithm 2 can be seen as an instance of the algorithm presented in Tropp et al. [2017] with specific choices of various tuning parameters optimized for our application. For more on this perspective, see Appendix E.2.

# 6 Experiments

**Algorithm 1 on real data.** We begin by confirming the accuracy and speed of Algorithm 1 on real data compared to both exact CV and existing ACV methods. We apply logistic regression to two datasets (p53 and `rcv1`) and Poisson regression to one dataset (`blog`). p53 has a size of roughly $N = 8{,}000, D = 5{,}000$, and the remaining two have roughly $N = D = 20{,}000$; see Appendix G for more details. For all experiments we fix $\lambda = 5.0$. We choose this moderate value of $\lambda$ to make the underlying optimization problems sufficiently regular so that exact CV's runtime would still be reasonable for our larger experiments. In Appendix H, we show that these results are not sensitive to the particular value of $\lambda$. To further speed up computation of exact LOOCV, we only run over twenty randomly chosen datapoints. We report average percent error, $(1/20) \sum_{b=1}^{20} |x_b^T \operatorname{appx.} - x_b^T \hat{\theta}_{\backslash b}| / |x_b^T \hat{\theta}_{\backslash b}|$ for each exact ACV algorithm and the output of Algorithm 1. For the smaller dataset p53, the speedup of Algorithm 1 over exact $\mathrm{NS}_{\backslash n}$ or $\mathrm{IJ}_{\backslash n}$ is marginal; however, for the larger two datasets, our methods provide significant speedups: for the `blog` dataset, we estimate the runtime of full exact CV to be nearly ten months. By contrast, the runtime of $\mathrm{NS}_{\backslash n}$ is nearly five minutes, and the runtime of $\widetilde{\mathrm{NS}}_{\backslash n}$ is forty seconds. In general, the accuracy of $\widetilde{\mathrm{IJ}}_{\backslash n}$ closely mirrors the accuracy of $\mathrm{IJ}_{\backslash n}$, while the accuracy of $\widetilde{\mathrm{NS}}_{\backslash n}$ can be somewhat worse than that

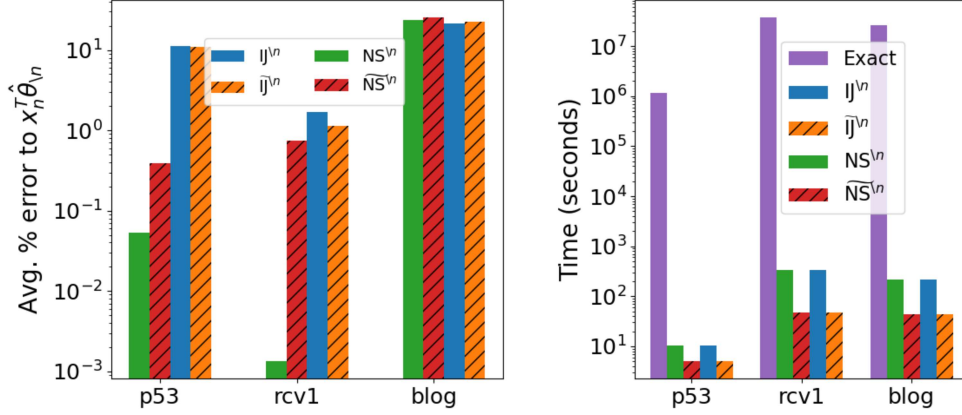

Figure 2: Experiments on real datasets. (*Left*): average percent error compared to exact CV on a subset of datapoints, $(1/20)\sum_{b=1}^{20}|x_b^T\mathrm{approx.} - x_b^T\hat{\theta}_{\backslash b}|/|x_b^T\hat{\theta}_{\backslash b}|$, where $\mathrm{approx.}$ denotes $\mathrm{NS}_{\backslash n}, \widetilde{\mathrm{NS}}_{\backslash n}, \mathrm{IJ}_{\backslash n}$, or $\widetilde{\mathrm{IJ}}_{\backslash n}$. (*Right*): ACV runtimes with exact CV runtimes for comparison. ACV runtimes are given for all $N$ datapoints. Exact CV runtimes are estimated runtimes for all $N$ datapoints.

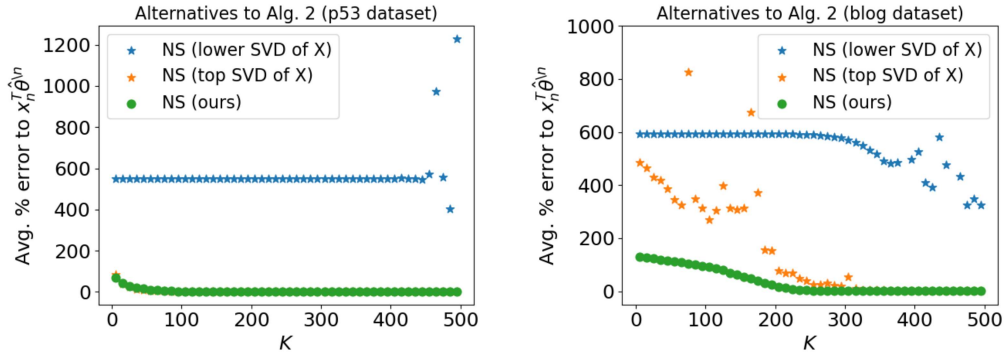

Figure 3: Comparisons to alternatives to Algorithm 1 for estimating $x_n^T\hat{\theta}_{\backslash n}$ via the NS approximation on small versions of two different real datasets: the p53 dataset (a logistic regression task) and the blog dataset (a Poisson regression task). See Appendix G for dataset details. For the p53 dataset (left), the results of our algorithm (green) and the top SVD of $X$ (orange) are visually indistinguishable. For the blog dataset (right), both the lower and upper SVD of $X$ obtain errors larger than the displayed scale; we cut off the vertical scale at 1,000% error so that finer details are visible.

of $\mathrm{NS}_{\backslash n}$ on the two logistic regression tasks; however, we note that in these cases, the error of $\widetilde{\mathrm{NS}}_{\backslash n}$ is still less than 1%.

**Alternatives for estimating $Q_n$.** Given our use of low-rank approximations to $H = X^T\mathrm{diag}\{\hat{D}_n^{(2)}\}_{n=1}^N X + \lambda I_D$ to approximate $Q_n$, one might first consider a more straightforward option before reaching for Algorithm 2. In particular, one might consider using principle components analysis, which is a common method for dimensionality reduction in generalized linear models. Here, this corresponds to taking the SVD of $X$ and then computing $H^{-1}$ using only the top-$K$ singular vectors and values. Additionally, as discussed in Section 5, one might consider using the *lower $K$* singular vectors and values given our use of $H^{-1}$. In Fig. 3, we study the performance of these options on smaller versions of the p53 and blog datasets. We see the value of our analysis in Propositions 2 and 3, as the lower singular vectors give an extremely poor approximation to CV. Further, we see the necessity of the truncation present in our final estimate $\widetilde{Q}_n$ in Proposition 4. In particular, recall that the $\mathrm{NS}_{\backslash n}$ approximation depends on $1/(1 - \hat{D}_n^{(2)}Q_n)$. Thus if, for any values of

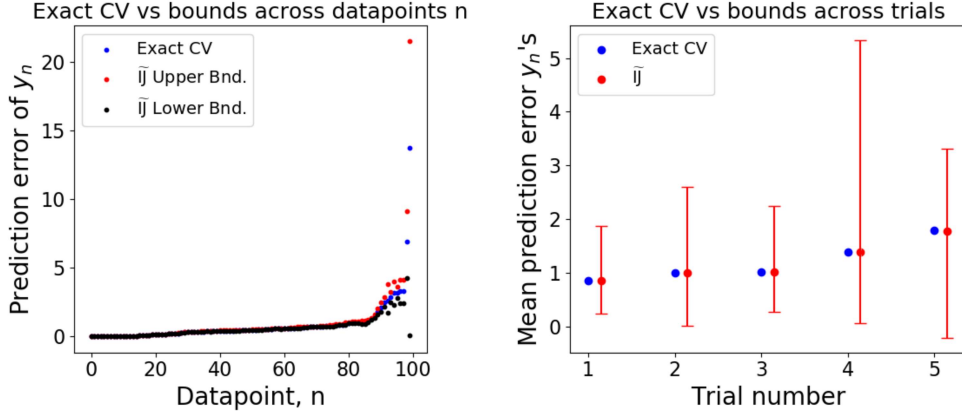

Figure 4: Error bounds implied by Theorem 1 on $\widetilde{\mathrm{IJ}}_{\backslash n}$'s estimate of out-of-sample error using squared loss, $\mathrm{Err}(x_n^T \theta, y_n) = (e^{x_n^T \theta} - y_n)^2$. (*Left*): Per datapoint error bounds. The bound is fairly loose for datapoints with larger squared loss, but tighter for points with lower squared loss. (*Right*): Five trials with estimates averaged across all $n$. We compute the upper (respectively, lower) error bars for $\widetilde{\mathrm{IJ}}_{\backslash n}$ by averaging the upper (respectively, lower) bounds. While our bounds overstate the difference between exact CV and $\widetilde{\mathrm{IJ}}_{\backslash n}$, they still give non-vacuous information on value of exact CV.

$n$ and $K$, the estimate of $\hat{D}_n^{(2)} Q_n$ passes near 1, the resulting estimate of $\mathrm{NS}_{\backslash n}$ will become unstable. We observe this instability for $K \leq \sim 300$ for the top-$K$ SVD of $X$ on the right of Fig. 3.

**Accuracy of error bounds.** We next empirically check the accuracy of the error bounds from Theorem 1. We generate a synthetic Poisson regression problem with i.i.d. covariates $x_{nd} \sim \mathcal{N}(0, 1)$ and $y_n \sim \mathrm{Poisson}(e^{x_n^T \theta^*})$, where $\theta^* \in \mathbb{R}^D$ is a true parameter with i.i.d. $\mathcal{N}(0, 1)$ entries. We generate a dataset of size $N = 800$ and $D = 500$ with covariates of approximate rank 50. To speed up the runtime of exact CV, we choose a moderate value of $\lambda = 1.0$. In Fig. 4, we illustrate the utility of the bounds from Theorem 1 by estimating the out-of-sample loss with $\mathrm{Err}(x_n^T \theta, y_n) = (e^{x_n^T \theta} - y_n)^2$. Across five trials, we show the results of exact LOOCV, our estimates provided by $\widetilde{\mathrm{IJ}}_{\backslash n}$, and the bounds on the error of $\widetilde{\mathrm{IJ}}_{\backslash n}$ given by Theorem 1. While our error bars in Fig. 4 tend to overestimate the difference between $\widetilde{\mathrm{IJ}}_{\backslash n}$ and exact CV, they typically provide upper bounds on exact CV on the order of the exact CV estimate itself. In some cases, we have observed that the error bars can overestimate exact CV by many orders of magnitude (see Appendix F), but this failure is usually due to one or two datapoints $n$ for which the bound is vacuously large. As these failure cases are easy to spot by inspection, a simple fix is to resort to exact CV just for these datapoints.

# 7 Conclusions

We provide an algorithm to approximate CV accurately and quickly in high-dimensional GLMs with ALR structure. Additionally, we provide quickly computable upper bounds on the error of our algorithm. We see two major directions for future work. First, while our theory and experiments focus on ACV for model assessment, the recent work of Wilson et al. [2020] has provided theoretical results on ACV for model selection (e.g. choosing $\lambda$). It would be interesting to see how dimensionality and ALR data plays a role in this setting. Second, as noted in the introduction, we hope that the results here will provide a springboard for studying ALR structure in models beyond GLMs and CV schemes beyond LOOCV.

**Broader Impact**

In general, we feel that work assessing the accuracy of machine learning models will have a positive impact on society. As machine learning is deployed in areas in which mistakes could have adverse effect on peoples' lives, it is important that we understand the error rate of such decisions before deployment. On the other hand, machine learning models can (and are) used for harm and the methods in this paper may assist in the development in such models. Additionally, there is always a risk in introducing any sort of approximation, as it may fail silently and unexpectedly in practice; e.g., our approximations might incorrectly lead a practitioner to conclude that their machine learning model has very small error when the opposite is in fact true. While we believe the computable upper bounds provided here somewhat mitigate this issue, we still remain cautious (though optimistic) about applying ACV methods in practice. Finally, we note that an implicit assumption throughout our work is that computing exact CV is something we want; that is, exact CV provides a good estimate of out-of-sample error. While this seems to be generally true, this does add another failure mode to our algorithm. In particular, even if we provide an accurate approximation to exact CV, it may be that exact CV itself is misleading.

**Acknowledgements**

The authors thank Zachary Frangella for helpful conversations. WS and TB were supported by the CSAIL-MSR Trustworthy AI Initiative, an NSF CAREER Award, an ARO Young Investigator Program Award, ONR Award N00014-17-1-2072, and Amazon. MU was supported by NSF Award IIS-1943131, the ONR Young Investigator Program, and DARPA Award FA8750-17-2-0101. The Broderick Group is also supported by the Sloan Foundation, ARPA-E, United States Department of the Air Force, and MIT Lincoln Laboratory.

## Footnotes

[1] In practice, for numerical stability, we compute a factorization of $H$ so that $H^{-1} x_n$ can be quickly evaluated for all $n$. However, for brevity, we refer to computation of the inverse of $H$ throughout.

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
