[Supplementary Material]

# A  Derivation of $x_n^T \mathrm{NS}_{\backslash n}$ and $x_n^T \mathrm{IJ}_{\backslash n}$

Here, we derive the expressions for $x_n^T \mathrm{NS}_{\backslash n}$ and $x_n^T \mathrm{IJ}_{\backslash n}$ given in Eqs. (3) and (4). We recall from previous work (e.g., see Stephenson and Broderick [2020, Appendix C] for a summary) that the LOOCV parameter estimates given by the Newton step and infinitesimal jackknife approximations are given by

$$\hat{\theta}_{\backslash n} \approx \mathrm{NS}_{\backslash n} := \hat{\theta} + \frac{1}{N} \left( \sum_{m:\, m \neq n}^{N} \hat{D}_n^{(2)} x_m x_m^T + \lambda I_D \right)^{-1} \hat{D}_n^{(1)} x_n \tag{14}$$

$$\hat{\theta}_{\backslash n} \approx \mathrm{IJ}_{\backslash n} := \hat{\theta} + \frac{1}{N} \left( \sum_{n=1}^{N} \hat{D}_n^{(2)} x_n x_n^T \lambda I_D \right)^{-1} \hat{D}_n^{(1)} x_n. \tag{15}$$

Taking the inner product of $\mathrm{IJ}_{\backslash n}$ with $x_n$ immediately gives Eq. (4). To derive Eq. (3), define $H := \sum_n \hat{D}_n^{(2)} x_n x_n^T + \lambda I_D$ and note that we can rewrite $\mathrm{NS}_{\backslash n}$ using the Sherman-Morrison formula:

$$\mathrm{NS}_{\backslash n} = \hat{\theta} + \frac{1}{N} \left[ \hat{D}_n^{(1)} H^{-1} x_n + \hat{D}_n^{(1)} \hat{D}_n^{(2)} \frac{H^{-1} x_n x_n^T H^{-1}}{1 - \hat{D}_n^{(2)} Q_n} x_n \right].$$

Taking the inner product with $x_n$ and reorganizing gives:

$$x_n^T \mathrm{NS}_{\backslash n} = x_n^T \hat{\theta} + \frac{\hat{D}_n^{(1)}}{N} \left[ \frac{Q_n - \hat{D}_n^{(2)} Q_n^2}{1 - \hat{D}_n^{(2)} Q_n} + \frac{\hat{D}_n^{(2)} Q_n^2}{1 - \hat{D}_n^{(2)} Q_n} \right] = x_n^T \hat{\theta} + \frac{\hat{D}_n^{(1)}}{N} \frac{Q_n}{1 - \hat{D}_n^{(2)} Q_n}.$$

# B  Comparison to existing Hessian inverse approximation

We note that two previous works have used inverse Hessian approximations for applications similar to ACV. Koh and Liang [2017] use influence functions to estimate behavior of black box models, and Lorraine et al. [2020] use the implicit function theorem to optimize model hyperparameters. In both papers, the authors need to multiply an inverse Hessian by a gradient. To deal with the high dimensional expense associated with this matrix inverse, both sets of authors use the method of Agarwal et al. [2017], who propose a stochastic approximation to the *Neumann series*. The Neumann series writes the inverse of a matrix $H$ with operator norm $\|H\|_{op} < 1$ as:

$$H^{-1} = \sum_{k=0}^{\infty} (I - H)^k.$$

The observation of Agarwal et al. [2017] is that this series can be written recursively, as well as estimated stochastically if one has random variables $A_s$ with $\mathbb{E}[A_s] = H$. In the general case of empirical risk minimization with an objective of $(1/N) \sum_{n=1}^{N} f_n(\theta)$, Agarwal et al. [2017] propose using $A_s = \nabla^2 f_s(\theta)$ for some $s \in [N]$ chosen uniformly at random. In the GLM setting we are interested in here, we choose an index $s \in [N]$ uniformly at random and set $A_s = \hat{D}_n^{(2)} x_s x_s^T + (\lambda/N) I_D$. Then, for $s = 1, \ldots, S$, we follow Agarwal et al. [2017] to recursively define:

$$H^{-1} \approx \bar{H}_s^{-1} := I_D + (I - A_s) \bar{H}_{s-1}^{-1}.$$

The final recommendation of Agarwal et al. [2017] is to repeat this process $M$ times and average the results. We thus have two hyperparameters to choose: $S$ and $M$.

To test out the Agarwal et al. [2017] approximation against our approximation in Algorithm 1, we generate Poisson regression datasets of increasing sizes $N$ and $D$. We generate approximately low-rank covariates $x_n$ by drawing $x_{nd} \sim N(0,1)$ for $d = 1, \ldots, 1{,}000$ and $x_{nd} \sim N(0, 0.01)$ for $d = 1{,}001, \ldots, D$; for our dataset with $D = 40$, we follow the same procedure but with $R = 20$ instead. For each dataset, we compute $\mathrm{IJ}_{\backslash n}$, as well as our approximation $\widetilde{\mathrm{IJ}}_{\backslash n}$ from Algorithm 1. We run Algorithm 1 for $K = 1, 100, 200, \ldots, D$ and run the stochastic Neumann series approximation with all combinations of $M \in \{2, 5\}$ and $S \in \{1, 5, 10, 15, \ldots, 200\}$. We measure the accuracy of

Figure 5: Experiment from Appendix B. Across four different dataset sizes, using the Neumann series approximation (orange) does not show any noticeable improvement on the time scale of running our approximation (green) for all possible values of $K$.

all approximations as percent error to exact CV ($x_n^T \hat{\theta}_{\backslash n}$). We show in Fig. 5 that our approximation has far improved error in far less time. Notably, this phenomenon becomes more pronounced as the dimension gets higher; while spending more computation on the Neumann series approximation does noticeably decrease the error for the $N = 80, D = 40$ case, we see that as soon as we step into even moderate dimensions ($D$ in the thousands), spending more computation on the Neumann approximation does not noticeably decrease the error. In fact, in the three lowest-dimensional experiments here, the dimension is so low that exactly computing $H^{-1}$ via a Cholesky decomposition is the fastest method.

We also notice that in the $N = 80, D = 40$ experiment, $\widetilde{IJ}_{\backslash n}$ is a better approximation of exact CV than is $IJ_{\backslash n}$ for intermediate values of $K$ (i.e. some orange dots sit below the large green dot). We note that we have observed this behavior in a variety of synthetic and real-data experiments. We do not currently have further insight into this phenomenon and leave its investigation for future work.

## C  Previous ACV theory

We briefly review pre-existing theoretical results on the accuracy of ACV. Theoretical results for the accuracy of $IJ_{\backslash n}$ are given by Giordano et al. [2019], Koh et al. [2019], Wilson et al. [2020]. Giordano et al. [2019] give a $O(1/N^2)$ error bound for unregularized problems, which Stephenson and Broderick [2020, Proposition 2] extends to regularized prolems; however, in our GLM case here, both results require the covariates and parameter space to be bounded. Koh et al. [2019] give a similar bound, but require the Hessian to be Lipschitz, and their bounds rely on the inverse of the minimum singular value of $H$, making them unsuited for describing the low rank case of interest here. The bounds of Wilson et al. [2020] are close to our bounds in Lemma 2. The difference to our work is that Wilson et al. [2020] consider generic (i.e. not just GLM) models, but also require a Lipschitz assumption on the Hessian. We specialize to GLMs, avoid the Lipschitz assumption by noting that it only need hold locally, and provide fully computable bounds.

Various theoretical guarantees also exist for the quality of $\text{NS}_{\backslash n}$ from Eq. (3). Rad and Maleki [2020] show that the error $\|\text{NS}_{\backslash n} - \hat{\theta}_{\backslash n}\|_2$ is $o(1/N)$ as $N \to \infty$ and give conditions under which the error is a much slower $O(1/\sqrt{N})$ as both $N, D \to \infty$ with $N/D$ converging to a constant. Beirami et al. [2017] show that the error is $O(1/N^2)$, but require fairly strict assumptions (namely, boundedness of the covariates and parameter space). Koh et al. [2019], Wilson et al. [2020] provide what seem to be the most interpretable bounds, but, as is the case for $\text{IJ}_{\backslash n}$, both require a Lipschitz assumption on the Hessian and the results of Koh et al. [2019] depend on the lowest singular value of the Hessian.

# D Proofs from Sections 3 and 4

## D.1 Proving accuracy of $\text{NS}_{\backslash n}$ and $\text{IJ}_{\backslash n}$ under exact low-rank data (Lemma 1)

Here, we prove that, when the covariate matrix is exactly rank $R \ll D$, the accuracy of $\text{NS}_{\backslash n}$ and $\text{IJ}_{\backslash n}$ behaves exactly as in a dimension $R \ll D$ problem. Let $X = U\Sigma V$ be the singular value decomposition of $X$, where $\Sigma$ is a diagonal matrix with only $R$ non-zero entries; let $V_{:R} \in \mathbb{R}^{D \times R}$ be the right singular vectors of $X$ corresponding to these $R$ non-zero singular values. We define the restricted, $R$-dimensional problem with covariates $\tilde{x}_n := V_{:R}^T x_n$ as:

$$\hat{\phi} := \arg\min_{\phi \in \mathbb{R}^R} \frac{1}{N} \sum_{n=1}^N f(\tilde{x}_n^T \phi) + \frac{\lambda}{2} \|\phi\|_2^2. \tag{16}$$

Let $\hat{\phi}_{\backslash n}$ be the solution to the leave-one-out version of this problem and $\text{RIJ}_{\backslash n}$ and $\text{RNS}_{\backslash n}$ the application of Eqs. (3) and (4) to this problem. We then have the following proposition, which implies the statement of Lemma 1.

**Proposition 5** (Generalization of Lemma 1). *The following hold for all datapoints $n$:*

$$x_n^T \hat{\theta}_{\backslash n} = \tilde{x}_n^T \hat{\phi}_{\backslash n}$$
$$x_n^T \text{IJ}_{\backslash n} = \tilde{x}_n^T \text{RIJ}_{\backslash n}$$
$$x_n^T \text{NS}_{\backslash n} = \tilde{x}_n^T \text{RNS}_{\backslash n}.$$

*In particular, $|x_n^T \text{NS}_{\backslash n} - x_n^T \hat{\theta}_{\backslash n}| = |\tilde{x}_n^T \text{RNS}_{\backslash n} - \tilde{x}_n^T \hat{\phi}_{\backslash n}|$ and $|x_n^T \text{IJ}_{\backslash n} - x_n^T \hat{\theta}_{\backslash n}| = |\tilde{x}_n^T \text{RIJ}_{\backslash n} - \tilde{x}_n^T \hat{\phi}_{\backslash n}|$, as claimed in Lemma 1.*

*Proof.* First, note that if $\hat{\phi}$ is an optimum of Eq. (16), then $(1/N) \sum_n \hat{D}_n^{(1)} V_{:R}^T x_n + \lambda \hat{\phi} = 0$. As $V_{:R} V_{:R}^T x_n = x_n$, we have that $\hat{\theta} = V_{:R} \hat{\phi}$ is optimal for the full, $D$-dimensional, problem. This implies that $\hat{\phi} = V_{:R}^T \hat{\theta}$, and thus $x_n^T \hat{\theta} = \tilde{x}_n^T \hat{\phi}$. The same reasoning shows that $x_n^T \hat{\theta}_{\backslash n} = \tilde{x}_n^T \hat{\phi}_{\backslash n}$.

Now, notice that the Hessian of the restricted problem, $H_R$, is given by $H_R = (1/N) \sum_n V_{:R}^T x_n x_n^T V_{:R} \hat{D}_n^{(2)} + \lambda I_R \implies H_R^{-1} = V_{:R}^T H^{-1} V_{:R}$, where the $\hat{D}_n^{(2)}$ are evaluated at $\tilde{x}_n^T \hat{\phi} = x_n^T \hat{\theta}$. Also, the gradients of the restricted problem are given by $\nabla_\phi f(\tilde{x}_n^T \hat{\phi}, y_n) = \hat{D}_n^{(1)} V_{:R}^T x_n$. Thus the restricted IJ is:

$$\text{RIJ}_{\backslash n} = \hat{\phi} + H_R^{-1} V_{:R}^T x_n \hat{D}_n^{(1)} = V_{:R}^T \left( \hat{\theta} + H^{-1} x_n \hat{D}_n^{(1)} \right) = V_{:R}^T \text{IJ}_{\backslash n}.$$

Thus, we have $\tilde{x}_n^T \text{RIJ}_{\backslash n} = x_n^T V_{:R} V_{:R}^T \text{IJ}_{\backslash n} = (V_{:R} V_{:R}^T x_n)^T \text{IJ}_{\backslash n}$. Using $V_{:R} V_{:R}^T x_n = x_n$, we have that $\tilde{x}_n^T \text{RIJ}_{\backslash n} = x_n^T \text{IJ}_{\backslash n}$. The proof that $\tilde{x}_n^T \text{RNS}_{\backslash n} = x_n^T \text{NS}_{\backslash n}$ is identical. $\square$

## D.2 Proving accuracy of $\text{NS}_{\backslash n}$ and $\text{IJ}_{\backslash n}$ under ALR data (Lemma 2)

We will first need a few lemmas relating to how the exact solutions $\hat{\theta}_{\backslash n}$ and $\hat{\theta}$ vary as we leave datapoints out and move from exactly low-rank to ALR. We start by bounding $\|\hat{\theta} - \hat{\theta}_{\backslash n}\|_2$; this result and its proof are from Wilson et al. [2020, Lemma 16] specialized to our GLM context.

**Lemma 3.** *Assume that $\lambda > 0$. Then:*

$$\left\| \hat{\theta} - \hat{\theta}_{\backslash n} \right\|_2 \leq \frac{1}{N\lambda} |\hat{D}_n^{(1)}| \, \|x_n\|_2. \tag{17}$$

*Proof.* Let $F^{\backslash n}$ be the leave-one-out objective, $F^{\backslash n}(\theta) = (1/N)\sum_{m:\,m\neq n} f(x_m^T\theta, y_m) + (\lambda/2)\|\theta\|_2^2$. As $F^{\backslash n}$ is strongly convex with parameter $\lambda$, we have:

$$\lambda \left\|\hat{\theta} - \hat{\theta}_{\backslash n}\right\|_2^2 \leq \langle \hat{\theta} - \hat{\theta}_{\backslash n}, \nabla F^{\backslash n}(\hat{\theta}) - \nabla F^{\backslash n}(\hat{\theta}_{\backslash n})\rangle.$$

Now, use the fact that $\nabla F^{\backslash n}(\hat{\theta}_{\backslash n}) = \nabla F(\hat{\theta}) = 0$ and then that $F^{\backslash n} - F = (1/N)f(x_n^T\theta)$ to get:

$$= \langle \hat{\theta} - \hat{\theta}_{\backslash n}, \nabla F^{\backslash n}(\hat{\theta}) - \nabla F(\hat{\theta})\rangle = \langle \hat{\theta} - \hat{\theta}_{\backslash n}, \nabla f(x_n^T\hat{\theta})\rangle$$
$$\leq \left\|\hat{\theta} - \hat{\theta}_{\backslash n}\right\|_2 |\hat{D}_n^{(1)}| \|x_n\|_2.$$

$\square$

We will need a bit more notation to discuss the ALR and exactly low-rank versions of the same problem. Suppose we have a $N \times D$ covariate matrix $X$ that is exactly low-rank (ELR) with rows $x_{n,ELR} \in \mathbb{R}^D$. Then, suppose we form some approximately low-rank (ALR) covariate matrix by adding $\varepsilon_n \in \mathbb{R}^D$ to all $x_n$ such that $X\varepsilon_n = 0$ for all $\varepsilon_n$. Let $x_{n,ALR}$ be the rows of this ALR matrix. Let $\hat{\theta}_{ELR}$ be the fit with the ELR data and $\hat{\theta}_{ALR}$ the fit with the ALR data. Finally, define the scalar derivatives:

$$\hat{D}_{n,ELR}^{(1)}(\theta) := \left.\frac{df(z, y_n)}{dz}\right|_{z=\langle x_{n,ELR},\theta\rangle}$$
$$\hat{D}_{n,ALR}^{(1)}(\theta) := \left.\frac{df(z, y_n)}{dz}\right|_{z=\langle x_{n,ALR},\theta\rangle}$$

We can now give an upper bound on the difference between the ELR and ALR fits $\|\hat{\theta}_{ELR} - \hat{\theta}_{ALR}\|_2$. Our bound will imply that the $\hat{\theta}_{ALR}$ is a continuous function of the $\varepsilon_n$, which in turn are continuous functions of the singular values of the ALR covariate matrix.

**Lemma 4.** *Assume $\lambda > 0$. We have:*

$$\left\|\hat{\theta}_{ELR} - \hat{\theta}_{ALR}\right\|_2 \leq \frac{1}{N\lambda} \left\|\sum_{n=1}^N \hat{D}_{n,ELR}^{(1)}(\hat{\theta}_{ELR})\varepsilon_n\right\|_2$$

*In particular, $\hat{\theta}_{ALR}$ is a continuous function of the $\varepsilon_n$ around $\varepsilon_1, \ldots, \varepsilon_N = 0$.*

*Proof.* Denote the ALR objective by $F_{ALR}(\theta) = (1/N)\sum_n f(x_{n,ALR}^T\theta) + \lambda\|\theta\|_2^2$. Then, via a Taylor expansion of its gradient around $\hat{\theta}_{ALR}$:

$$\nabla_\theta F_{ALR}(\hat{\theta}_{ELR}) = \nabla_\theta F_{ALR}(\hat{\theta}_{ALR}) + \nabla_\theta^2 F_{ALR}(\tilde{\theta})(\hat{\theta}_{ELR} - \hat{\theta}_{ALR}),$$

where $\tilde{\theta} \in \mathbb{R}^D$ satisfies $\tilde{\theta}_d = (1-s_d)\theta_{ALR,d} + s_d\theta_{ELR,d}$ for some $s_d \in [0,1]$ for each $d = 1, \ldots, D$. Via strong convexity and $\nabla_\theta F_{ALR}(\hat{\theta}_{ALR}) = 0$, we have:

$$\left\|\hat{\theta}_{ELR} - \hat{\theta}_{ALR}\right\|_2 \leq \frac{1}{\lambda}\left\|\nabla_\theta F_{ALR}(\hat{\theta}_{ELR})\right\|_2.$$

Now, note that the gradient on the right hand side of this equation is equal to

$$\nabla_\theta F_{ALR}(\hat{\theta}_{ELR}) = \frac{1}{N}\sum_{n=1}^N \hat{D}_{n,ALR}^{(1)}(\hat{\theta}_{ELR})x_{n,ELR} + \frac{1}{N}\sum_{n=1}^N \hat{D}_{n,ALR}^{(1)}(\hat{\theta}_{ELR})\varepsilon_n + \lambda\hat{\theta}_{ELR}. \quad (18)$$

By the optimality of $\hat{\theta}_{ELR}$ for the exactly low-rank problem, we must have that $\langle \varepsilon_n, \hat{\theta}_{ELR}\rangle = 0$ for all $n$; in particular, this implies that $\langle x_{n,ELR}, \hat{\theta}_{ELR}\rangle = \langle x_{n,ALR}, \hat{\theta}_{ELR}\rangle$, which in turn implies $\hat{D}_{n,ALR}^{(1)}(\hat{\theta}_{ELR}) = \hat{D}_{n,ELR}^{(1)}(\hat{\theta}_{ELR})$ for all $n$. Also by the optimality of $\hat{\theta}_{ELR}$, we have $(1/N)\sum_n \hat{D}_{n,ELR}^{(1)}(\hat{\theta}_{ELR})x_n + \lambda\hat{\theta}_{ELR} = 0$. Thus we have that Eq. (18) reads:

$$\nabla_\theta F_{ALR}(\hat{\theta}_{ELR}) = \frac{1}{N}\sum_{n=1}^N \hat{D}_{n,ELR}^{(1)}(\hat{\theta}_{ELR})\varepsilon_n,$$

which completes the proof.

$\square$

We now restate and prove Lemma 2.

**Lemma 2.** *Assume that $\lambda > 0$ and recall the definition of $L_n$ from Eq. (8). Then, for all $n$:*

$$|x_n^T \text{NS}_{\backslash n} - x_n^T \hat{\theta}_{\backslash n}| \leq \frac{L_n}{N^2 \lambda^3} |\hat{D}_n^{(1)}|^2 \, \|x_n\|_2^3 \qquad (19)$$

$$|x_n^T \text{IJ}_{\backslash n} - x_n^T \hat{\theta}_{\backslash n}| \leq \frac{L_n}{N^2 \lambda^3} |\hat{D}_n^{(1)}|^2 \, \|x_n\|_2^3 + \frac{1}{N^2 \lambda^2} |\hat{D}_n^{(1)}| |\hat{D}_n^{(2)}| \, \|x_n\|_2^4 \, . \qquad (20)$$

*Furthermore, these bounds continuously decay as the data move from exactly to approximately low rank in that they are continuous in the singular values of $X$.*

*Proof.* The proof of Eqs. (19) and (20) strongly resembles the proof of Wilson et al. [2020, Lemma 17] specialized to our current context. We first prove Eq. (19). We begin by applying the Cauchy-Schwarz inequality to get:

$$|x_n^T \text{NS}_{\backslash n} - x_n^T \hat{\theta}_{\backslash n}| \leq \|x_n\|_2 \left\| \text{NS}_{\backslash n} - \hat{\theta}_{\backslash n} \right\|_2 \, .$$

The remainder of our proof focuses on bounding $\|\text{NS}_{\backslash n} - \hat{\theta}_{\backslash n}\|_2$. Let $\widetilde{F^{\backslash n}}$ be the second order Taylor expansion of $F^{\backslash n}$ around $\hat{\theta}$; then $\text{NS}_{\backslash n}$ is the minimizer of $\widetilde{F^{\backslash n}}$. By the strong convexity of $\widetilde{F^{\backslash n}}$:

$$\lambda \left\| \hat{\theta}_{\backslash n} - \text{NS}_{\backslash n} \right\|_2^2 \leq \langle \text{NS}_{\backslash n} - \hat{\theta}_{\backslash n}, \nabla \widetilde{F^{\backslash n}}(\text{NS}_{\backslash n}) - \nabla \widetilde{F^{\backslash n}}(\hat{\theta}_{\backslash n}) \rangle \qquad (21)$$

$$= \langle \text{NS}_{\backslash n} - \hat{\theta}_{\backslash n}, \nabla F^{\backslash n}(\hat{\theta}_{\backslash n}) - \nabla \widetilde{F^{\backslash n}}(\hat{\theta}_{\backslash n}) \rangle \qquad (22)$$

Now the goal is to bound this quantity as the remainder in a Taylor expansion. To this end, define $r(\theta) := \langle \hat{\theta}_{\backslash n} - \text{NS}_{\backslash n}, \nabla F^{\backslash n}(\theta) \rangle$. To apply Taylor's theorem with integral remainder, define $g(t) := r((1-t)\hat{\theta} + t\hat{\theta}_{\backslash n})$ for $t \in [0,1]$. Then, by a zeroth order Taylor expansion:

$$g(1) = g(0) + g'(0) + \int_0^1 (g'(s) - g'(0)) \, ds.$$

Putting in the values of $g$ and its derivatives:

$$\langle \hat{\theta}_{\backslash n} - \text{NS}_{\backslash n}, \nabla F^{\backslash n}(\hat{\theta}_{\backslash n}) \rangle = \langle \hat{\theta}_{\backslash n} - \text{NS}_{\backslash n}, \nabla F^{\backslash n}(\hat{\theta}) \rangle + \langle \hat{\theta}_{\backslash n} - \text{NS}_{\backslash n}, \nabla^2 F^{\backslash n}(\hat{\theta})(\hat{\theta}_{\backslash n} - \hat{\theta}) \rangle$$

$$+ \int_0^1 \langle \hat{\theta}_{\backslash n} - \text{NS}_{\backslash n}, \left( \nabla^2 F^{\backslash n}((1-s)\hat{\theta} + s\hat{\theta}_{\backslash n}) - \nabla^2 F^{\backslash n}(\hat{\theta}) \right) (\hat{\theta}_{\backslash n} - \hat{\theta}) \rangle ds$$

Now, subtracting the first two terms on the right hand side from the left, we get can identify the left with Eq. (22). Thus, Eq. (22) is equal to:

$$= \int_0^1 \langle \hat{\theta}_{\backslash n} - \text{NS}_{\backslash n}, \left( \nabla^2 F^{\backslash n}((1-s)\hat{\theta} + s\hat{\theta}_{\backslash n}) - \nabla^2 F^{\backslash n}(\hat{\theta}) \right) (\hat{\theta}_{\backslash n} - \hat{\theta}) \rangle ds.$$

We can upper bound this by taking an absolute value, then applying the triangle inequality and Cauchy-Schwarz to get

$$\leq \left\| \hat{\theta}_{\backslash n} - \text{NS}_{\backslash n} \right\|_2 \left\| \hat{\theta}_{\backslash n} - \hat{\theta} \right\| \int_0^1 \left\| \left( \nabla^2 F^{\backslash n}((1-s)\hat{\theta} + s\hat{\theta}_{\backslash n}) - \nabla^2 F^{\backslash n}(\hat{\theta}) \right) \right\|_{op} ds. \qquad (23)$$

Using the fact that, on the line segment $(1-s)\hat{\theta} + s\hat{\theta}_{\backslash n}$, the $\hat{D}_n^{(2)}$ are lipschitz with constant $C_n$:

$$C_n := \max_{s = in[0,1]} \left| \hat{D}_n^{(3)} \left( (1-s)\hat{\theta} + s\hat{\theta}_{\backslash n} \right) \right|,$$

we can upper bound the integrand by:

$$\left\| \left( \nabla^2 F^{\backslash n}((1-s)\hat{\theta} + s\hat{\theta}_{\backslash n}) - \nabla^2 F^{\backslash n}(\hat{\theta}) \right) \right\|_{op}$$

$$= \frac{1}{N} \left\| \sum_{m \neq n} \left( \hat{D}_n^{(2)}((1-s)\hat{\theta} + s\hat{\theta}_{\backslash n}) - \hat{D}_n^{(2)}(\hat{\theta}) \right) x_m x_m^T \right\|_{op}$$

$$\leq \frac{C_n \left\| \hat{\theta}_{\backslash n} - \hat{\theta} \right\|_2}{N} \sum_{m \neq n} \|x_m\|_2^2 \, .$$

Putting this into Eq. (23) and using Lemma 3 gives the result Eq. (19) with $L_n = C_n/N \sum_{m \neq n} \|x_m\|_2^2$.

Now Eq. (20) follows from the triangle inequality $\|\mathrm{IJ}_{\backslash n} - \hat\theta_{\backslash n}\|_2 \leq \|\mathrm{NS}_{\backslash n} - \hat\theta_{\backslash n}\|_2 + \|\mathrm{IJ}_{\backslash n} - NS\|_2$. The bound on $\|\mathrm{IJ}_{\backslash n} - \mathrm{NS}_{\backslash n}\|_2$ follows from Wilson et al. [2020, Lemma 20].

Finally, the continuity of the bounds in Eqs. (19) and (20) follows from Lemma 4. In particular, the $\hat{D}_n^{(1)}$, $\hat{D}_n^{(2)}$, and $\hat{D}_n^{(3)}$ in both bounds are evaluated at $\hat\theta_{ALR}$, which is shown to be a continuous function of the $\varepsilon_n$ in Lemma 4. The $\varepsilon_n$ are, in turn, continuous functions of the lower singular values of the covariate matrix. $\qquad\square$

### D.3 Proof of Theorem 1

*Proof.* We first note that the runtime claim is immediate, as Algorithm 2 runs in $O(NDK + K^3)$ time. That the bounds are computable in $O(DK)$ time for each $n$ follows as all derivatives $\hat{D}_n^{(1)}$ and $\hat{D}_n^{(2)}$ need only the inner product of $x_n$ and $\hat\theta$, which takes $O(D)$ time. Each norm $\|x_n\|_2$ is computable in $O(D)$. For models for which the optimization problem in Proposition 1 can be quickly solved – such as Poisson or logistic regression – we need only to compute a bound on $\|\hat\theta_{\backslash n} - \hat\theta\|_2$, which we can do in $O(D)$ via Lemma 3. The only remaining quantity to compute is the $\eta_n$, which, by Proposition 2, is computed via a projection onto the orthogonal complement of a $K$-dimensional subspace. We can compute this projection in $O(DK)$. Thus our overall runtime is $O(DK)$ per datapoint.

To prove Eq. (7), we use the triangle inequality $|x_n^T \widetilde{\mathrm{IJ}}_{\backslash n} - x_n^T \hat\theta_{\backslash n}| \leq |x_n^T \mathrm{IJ}_{\backslash n} - x_n^T \hat\theta_{\backslash n}| + |x_n^T \mathrm{IJ}_{\backslash n} - x_n^T \widetilde{\mathrm{IJ}}_{\backslash n}|$. We upper bound the first term by using Lemma 2. For the latter, we note that $|x_n^T \mathrm{IJ}_{\backslash n} - x_n^T \widetilde{\mathrm{IJ}}_{\backslash n}| = |\hat{D}_n^{(1)}||Q_n - \tilde{Q}_n|$, which we can bound via the $\eta_n$ of Proposition 2. The proof for $\mathrm{NS}_{\backslash n}$ is similar. $\qquad\square$

### D.4 Proof of Corollary 1

*Proof.* Notice that $\Omega$ from Algorithm 2 captures a rank-$K$ subspace of the column span of $X$. The error bound $\eta_n$ is the norm of $x_n$ projected outside of this subspace divided by $\lambda$. Now, assume that we have $K \geq R$. Then, as the singular values $\sigma_d$ for $d = R+1, \ldots, D$ go to zero, the norm of any $x_n$ outside this subspace must also go to zero. Thus $\eta_n$ goes to zero. As $E_n$ is a continuous function of $\eta_n$, we also have $E_n \to 0$. $\qquad\square$

## E Proofs and discussion from Section 5

### E.1 Proofs

For convenience, we first restate each claimed result from the main text before giving its proof.

**Proposition 2.** *Let $\lambda > 0$ and suppose there is some subspace $\mathcal{B}$ on which $H$ and $\widetilde{H}$ exactly agree. Then $H^{-1}$ and $\widetilde{H}^{-1}$ agree exactly on the subspace $\mathcal{A} := H\mathcal{B}$, and for all $n = 1, \ldots, N$:*

$$|x_n^T \widetilde{H}^{-1} x_n - Q_n| \leq \frac{\left\| P_{\mathcal{A}}^{\perp} x_n \right\|_2^2}{\lambda}, \tag{24}$$

*where $P_{\mathcal{A}}^{\perp}$ denotes projection onto the orthogonal complement of $\mathcal{A}$.*

*Proof.* First, if $H$ and $\widetilde{H}$ agree on $\mathcal{B}$, then for $\mathcal{A} = H\mathcal{B} = \widetilde{H}\mathcal{B}$, we have $H^{-1}\mathcal{A} = \mathcal{B} = \widetilde{H}^{-1}\mathcal{A}$, as claimed. Then:

$$|Q_n - x_n^T \widetilde{H}^{-1} x_n| = |x_n^T H^{-1} x_n - x_n^T \widetilde{H}^{-1} x_n|$$
$$\leq |(P_{\mathcal{A}}^{\perp} x_n)(H^{-1} - \widetilde{H}^{-1})(P_{\mathcal{A}}^{\perp} x_n)|$$
$$\leq \left\| P_{\mathcal{A}}^{\perp} x_n \right\|_2^2 \left\| H^{-1} - \widetilde{H}^{-1} \right\|_{op, \mathcal{A}^{\perp}},$$

where $\|\cdot\|_{op,\mathcal{A}^\perp}$ is the operator norm of a matrix restricted to the subspace $\mathcal{A}^\perp$. On this subspace, the action of $\widetilde{H}^{-1}$ is $1/\lambda$ times the identity, whereas all eigenvalues of $H^{-1}$ are all between 0 and $1/\lambda$. Thus:

$$\left\|\widetilde{H}^{-1} - H^{-1}\right\|_{op,\mathcal{A}^\perp} = \max_{v\in\mathcal{A}^\perp,\,\|v\|_2=1}\left[v^T\widetilde{H}^{-1}v - v^T H^{-1}v\right]$$

$$= \max_{v\in\mathcal{A}^\perp,\,\|v\|_2=1}\left[\frac{1}{\lambda} - v^T H^{-1}v\right] \le \frac{1}{\lambda}.$$

$\square$

We next restate and proof Proposition 4.

**Proposition 4.** *The* $Q_n = x_n^T H^{-1}x_n$ *satisfy* $0 \le Q_n \le \|x_n\|_2^2/(\lambda + \hat{D}_n^{(2)}\|x_n\|_2^2)$. *Furthermore, letting* $\widetilde{Q}_n := \min\{x_n^T\widetilde{H}^{-1}x_n, \|x_n\|_2^2/(\lambda + \hat{D}_n^{(2)}\|x_n\|_2^2)\}$, *we have the error bound*

$$|\widetilde{Q}_n - Q_n| \le \min\left\{\frac{\left\|P_\mathcal{A}^\perp x_n\right\|_2^2}{\lambda}, \frac{\|x_n\|_2^2}{\lambda + \hat{D}_n^{(2)}\|x_n\|_2^2}\right\}. \qquad (25)$$

*Proof.* Let $b_n := \sqrt{\hat{D}_n^{(2)}}x_n$. Let $\{v_d\}_{d=1}^D$ be the eigenvectors of $H$ with eigenvalues $\{\gamma_d + \lambda\}_{d=1}^D$ with $\gamma_1 \ge \gamma_2 \ge \cdots \ge \gamma_D$. The quantity $b_n^T H^{-1}b_n$ is maximized if $b_n$ is parallel to $v_D$; in this case, $b_n^T H^{-1}b_n = \|b_n\|_2^2/(\gamma_D + \lambda)$. Now, recall that the $\gamma_d$ are the eigenvalues of $\sum_n b_n b_n^T$, meaning $\sum_n \langle b_n, v_D\rangle^2 = \gamma_D$. So, if $b_n$ is parallel to $v_D$, it must be that $\gamma_D \ge \|b_n\|_2^2$. Thus, $b_n^T H^{-1}b_n \le \|b_n\|_2^2/(\|b_n\|_2^2 + \lambda)$. Dividing by $\hat{D}_n^{(2)}$ gives that $Q_n = x_n^T H^{-1}x_n$ satisfies:

$$0 \le Q_n \le \frac{\|x_n\|_2^2}{\lambda + \hat{D}_n^{(2)}\|x_n\|_2^2}.$$

If we estimate $Q_n$ by the minimum of this upper bound and $x_n^T\widetilde{H}^{-1}x_n$, the error bound from Proposition 2 implies the error bound claimed here. $\square$

### E.2 Relation of Algorithm 2 to techniques from randomized linear algebra

As noted, our Algorithm 2 bears a resemblance to techniques from the randomized numerical linear algebra literature. Indeed, our inspiration for Algorithm 2 was the work of Tropp et al. [2017]. Tropp et al. [2017] propose a method to find a randomized top-$K$ eigendecomposition of a positive-semidefinite matrix $B$. Their method follows the basic steps of (1) produce a random orthonormal matrix $\Omega \in \mathbb{R}^{D\times(S+K)}$, where $S \ge 0$ is an *oversampling* parameter to ensure the stability of the estimated eigendecomposition, (2) compute the Nyström approximation of $B_{nys} \approx B$ using $\Omega$, and (3) compute the eigendecomposition of $B_{nys}$ and throw away the lowest $S$ eigenvalues.

Our Algorithm 2 can be seen as using this method of Tropp et al. [2017] to obtain a rank-$K$ decomposition of the matrix $B = (1/N)\sum_n \hat{D}_n^{(2)}x_n x_n^T$ with specific choices of $S$ and $\Omega$. First, we notice that $S = 0$ (i.e., no oversampling) is optimal in our application – the error bound of Proposition 2 decreases as the size of the subspace $\mathcal{A}$ increases. As $S > 0$ only decreases the size of this subspace, we see that our specific application is only hurt by oversampling. Next, while Tropp et al. [2017] recommend completely random matrices $\Omega$ for generic applications (e.g., the entries of $\Omega$ are i.i.d. $\mathcal{N}(0,1)$), we note that the results of Proposition 3 suggest that we can improve upon this choice. With the optimal choice of $S = 0$, we note that $\mathcal{A} = H\Omega$. In this case, Proposition 3 implies it is optimal to set $\Omega = H^{-1}V_{:K}$, where $V_{:K}$ are the top-$K$ right singular vectors of $X$. Algorithm 2 provides an approximation to this optimal choice.

We illustrate the various possible choices of $\Omega$, including i.i.d. $\mathcal{N}(0,1)$, in Fig. 6. We generate a synthetic Poisson regression problem with covariates $x_{nd} \overset{i.i.d.}{\sim} \mathcal{N}(0,1)$ and $y_n \sim \text{Poisson}(e^{x_n^T\theta^*})$, where $\theta^* \in \mathbb{R}^D$ is a true parameter with i.i.d. $\mathcal{N}(0,1)$ entries. We generate a dataset of size $N = 200$. The covariates have dimension $D = 150$ but are of rank 50. We compute $\widetilde{IJ}$ for various settings of $K$ and $\Omega$, as shown in Fig. 6. As suggested by the above discussion, we use no oversampling

Figure 6: Quality of approximation of $\mathrm{IJ}_{\setminus n}$ on a synthetic Poisson regression problem using the methods from Section 5. (*Left*): We show three options for the choice of the matrix $\Omega$. Blue shows the choice of $\Omega$ having orthonormal columns selected uniformly at random, orange the optimal choice of $\Omega$ from Proposition 3, and green our approximation to this optimal choice. Percent error $|\mathrm{IJ}_{\setminus n} - \widetilde{\mathrm{IJ}_{\setminus n}}|/|\mathrm{IJ}_{\setminus n}|$ is reported to give a sense of scale. (*Right*): Importance of Proposition 4 for approximating $Q_n$. We show two approximations along with our upper bounds on their error: (1) $Q_n \approx x_n^T \widetilde{H}^{-1} x_n$ and (2) our recommended $Q_n \approx \widetilde{Q}_n$ from Proposition 4. We report absolute error $|\mathrm{IJ}_{\setminus n} - \widetilde{\mathrm{IJ}_{\setminus n}}|$ so that both actual and estimated error can be plotted.

(i.e., $S = 0$). On the left, we see that using a diagonal approximation to $H^{-1}$ and a single subspace iteration gives a good approximation to the optimal setting of $\Omega$. On the right, we see the improvement made by use of the upper bound on $x_n^T H^{-1} x_n$ from Proposition 4.

### E.3 Implementation of Algorithm 2

As noted by Tropp et al. [2017], finding the decomposition of $B$ in Algorithm 2 as-written can result in numerical issues. Instead, Tropp et al. [2017] present a numerically stable version which we use in our experiments. For completeness, we state this implementation here, which relies on computing the Nyström approximation of the shifted matrix $B_\nu = B + \nu I_D$, for some small $\nu > 0$:

1. Construct the shifted matrix sketch $G_\nu := B\Omega + \nu\Omega$.
2. Form $C = \Omega^T G_\nu$.
3. Compute the Cholesky decomposition $C = \Gamma\Gamma^T$.
4. Compute $E = G_\nu \Gamma^{-1}$.
5. Compute the SVD $E = U\Sigma V^T$.
6. Return $U$ and $\Sigma^2 - \nu I$ as the approximate eigenvectors and eigenvalues of $B$.

## F  Error bound experiments

Here, we provide more details on our investigation of the error bounds of Theorem 1 from Fig. 4. In Section 6, we showed that, over five randomly generated synthetic datasets, our error bound on $x_n^T \widetilde{\mathrm{IJ}}_{\setminus n}$ implies upper bounds on out-of-sample error that are reasonably tight. However, we noted that these bounds can occasionally be vacuously loose. On the left of Appendix F, we show this is the case by repeating the experiment in Fig. 4 for an additional fifteen trials. While most trials have similar behavior to the first five, trial 16 finds an upper bound of the out-of-sample error that is too loose by two orders of magnitude. However, we note that this behavior is mostly due to two offending points $n$. Indeed, on the right of Appendix F, we show the same results having replaced the two largest bound values with those from exact CV.

Figure 7: Experiments from Appendix F. (*Left*): Error bounds can be vacuously large; for trial number 16, our bound exceeds exact CV by two orders of magnitude. (*Right*): By computing our bound for all $n$ and re-running exact CV for the two largest values, we obtain estimates that are much closer to exact CV.

## G   Real data experiments

Here we provide more details about the three real datasets used in Section 6.

1. The `p53` dataset is from Danziger et al. [2009, 2007, 2006]. The full dataset contains $D = 5{,}408$ features describing attributes of mutated p53 proteins. The task is to classify each protein as either "transcriptionally competent" or inactive. To keep the dimension high relative to the number of observations $N$, we subsampled $N = 8{,}000$ datapoints uniformly at random for our experiments here. We fix $K = 500$ to compute $\widetilde{Q}_n$ for both $x_n^T \widetilde{\text{IJ}}_{\setminus n}$ and $x_n^T \widetilde{\text{NS}}_{\setminus n}$.

2. The `rcv1` dataset is from Lewis et al. [2004]. The full dataset is of size $N = 20{,}242$ and $D = 47{,}236$. Each datapoint corresponds to a Reuters news article given one of four labels according to its subject: "Corporate/Industrial," "Economics," "Government/Social," and "Markets." We use a pre-processed binarized version from https://www.csie.ntu.edu.tw/~cjlin/libsvmtools/datasets/binary.html, which combines the first two categories into a "positive" label and the latter two into a "negative" label. We found running CV on the full dataset to be too computationally intensive, and instead formed a smaller dataset of size $N = D = 20{,}000$. The data matrix is highly sparse, so we chose our 20,000 dimensions by selecting the most common (i.e., least sparse) features. We then chose $N = 20{,}000$ datapoints by subsampling uniformly at random. We fix $K = 1{,}000$ in this experiment.

3. The `blog` dataset is from Buza [2014]. The base dataset contains $D = 280$ features about $N = 52{,}397$ blogs. Each feature represents a statistic about web traffic to the given blog over a 72 hour period. The task is to predict the number of unique visitors to the blog in the subsequent 24 hour period. We first generate a larger dataset by considering all possible pairwise features $x_{nd_1} x_{nd_2}$ for $d_1, d_2 \in \{1, \ldots, D\}$. The resulting problem has too high $N$ and $D$ to run exact CV on in a reasonable amount of time, so we again subsample to $N = 20{,}000$ and $D = 20{,}280$. We again choose the 20,000 least sparse parwise features and then add in the original 280 features. Finally, we choose our 20,000 datapoints uniformly at random.

## H   Sensitivity of results to $\lambda$

In our main real-data experiments in Section 6, we chose a moderate value of $\lambda = 5.0$ to speed up the convergence of exact CV. Here, we investigate how sensitive our results are to the choice of this parameter. We randomly select $N = 600, D = 400$ subsets of the `p53` and `blog` datasets. We exactly compute $Q_n$, as well as compute our approximation $\widetilde{Q}_n$ from Algorithm 2 for $K \in$

Figure 8: Experiment from Appendix H. We report average error in the estimate of $Q_n$, $(1/N)\sum_n |Q_n - \widetilde{Q}_n|/|Q_n|$ for both the p53 and blog datasets. We note that the errors when using $K = 300, 350, 400$ are visually indistinguishable from one another.

$\{100, 150, 200, 250, 300, 350, 400\}$. In Fig. 8, we see that the choice of $\lambda$ has only a mild effect on the results.