[Reviews · NeurIPS 2020]

Review 1

Summary and Contributions: This paper analyzes two kinds of approximate leave one out estimates for the case of generalized linear regression with low-rank features.

Strengths: Cross-validation in high dimensions is a relevant and important topic to the NeurIPS community. The main contribution of the paper is the upper bound on the error between the actual leave-one-out estimate and the two kinds of approximation.

Weaknesses: I think the significance of the results (maybe because of the delivery of the result) is below the threshold of acceptance. 1) The first weakness is that there is no discussion about whether the upper bound (mentioned in the strengths) is tight and when this upper bound implies consistency, i,e., the error goes to 0 under a certain limit. Note that the norm of the true signal, the scale of the feature matrix, and the best tuning parameter need to satisfy certain order conditions such that the problem becomes meaningful. 2) the second weakness is that when the feature matrix is in fact low rank. A common approach is to apply PCA and do feature selection first. Then, the authors should compare their results with prior works on the selected features. After response: I noticed corollary 1 and corollary 2. But these two corollaries together only cover the trivial case when sample size goes to infinity while the rank of feature matrix is bounded by constant. The discussion of what the upper bounds imply is not sufficient for me. Since this paper focuses on low-rank data, one may ask is the low-rank data only refers to a bounded rank case? If not, is the bound sharp under other cases e.g. eigenvalue of feature covariance matrix has polynomial decays vs exponential decays? Further, lambda is now treated as constant. However, we know in high dimension settings, depending on the model assumption, the optimal lambda can depend on sample size. Is the upper bound still sharp for the optimal lambda? Another example is that if we look at (6) or (9) and scale the features by constant C and lambda by constant C^2, then the estimates theta_n and theta hat should be scaled down by C, hence the left-hand side of the bound should remain the same order while the right-hand side of the bound is scaled according to C^(-3) L_n \cdot D^1(z). If we consider z as constant order as it is an estimate of y, then it seems that the right-hand side is scaled with C^(-1). So does it mean it is always good to scale the features by large constant and it is bad to scale the features with small constant? Or the setting is not realistic because we should consider the optimal lambda instead when scaling the features. In summary, as the major contribution of this paper, I think it is the authors' responsibility to help the reader understand what is implied by the main theorems and although the paper looks interesting, it is not clear to me whether the bounds are trivial or not. Regarding PCA arguments, I agree that the full model can be still interesting because the true signal can be large when the variance of the corresponding features is small. However, my concern was the results in the paper may not be accurate in this situation. More specifically, in the low-rank case considered in the corollary, when the small eigenvalues are diminishing to 0, their corresponding true signal coefficients need to be large (large here is vague because it depends on how many small eigenvalues) so that PCA losses information than the full model. It is not clear whether the bound is still sharp in the situation. Other reviewers mentioned the practical value of this paper. I am willing to raise my score to 5. But I think the empirical results alone is not enough evidence to convince me. On the theoretical side, the implication of the error bound itself is not fully discussed. p.s. I notice the authors try to address whether the bound is sharp for lambda<=1 with extra experiment. However, (maybe it is a typo) the range of log lambda is positive and thus lambda is still at least 1.

Correctness: The claims look correct to me. There is a minor point is that the statement of bound computation time in Theorem 1 is not exactly correct because L_n or M_n needs at least DN times to compute. But it only needs to be computed once and can be then reused for all n. I can pass on this.

Clarity: Except for the lack of discussion of their main result of upper bound, it is clear to follow the paper.

Relation to Prior Work: It could be a good idea to cite some leave-one-out paper written by Noureddine to illustrate the importance of leave-one-out estimators for prediction risk estimation. References are the following. The proofs of the results depend on the analysis of the leave one out estimate. Noureddine El Karoui, Derek Bean, Peter J. Bickel, Chinghway Lim, and Bin Yu. On robust regression with highdimensional predictors. Proceedings of the National Academy of Sciences, page 201307842, 2013. Noureddine El Karoui On the impact of predictor geometry on the performance on high-dimensional ridge-regularized generalized robust regression estimators. Probability Theory and Related Fields, 170(1-2):95–175, 2018.

Reproducibility: Yes

Additional Feedback:


Review 2

Summary and Contributions: The authors propose an extension to recently proposed approximate cross-validation (ACV) techniques by constructing a low-rank approximation to the Hessian, an essential element for the aforementioned ACV techniques. The authors demonstrate how to leverage this low-rank decomposition to accelerate existing ACV techniques, and obtain computable error bounds for the obtained estimates. The authors demonstrate good accuracy and significant speed-ups in empirical results.

Strengths: The authors present a compelling improvement for ACV techniques applied to dense estimators by leveraging potential low-rank structure in the dataset, and provide an analysis of its characteristics that can help guide its usage. Due to the importance of fast model tuning for practitioners, this is of fairly general interest to the community.

Weaknesses: A major obstacle to such a method being of use to practitioners is that it introduces another hyperparameter: indeed, as real data is very seldom low-rank, the rank of the approximation itself must be chosen in order to compute the ALR estimate. Given that the main use of ACV techniques is selecting model parameters, it is somewhat unfortunate that the proposed method is not fully automatic and requires selecting a parameter to be used.

Correctness: The derivations appear to be correct.

Clarity: The paper is clear and well-written.

Relation to Prior Work: Reference to prior work is adequate.

Reproducibility: Yes

Additional Feedback: This paper is a good contribution in the domain of ACV estimates, and proposes a compelling method to produce ACV estimates for dense estimators, particularly in the context of GLMs. I have a few technical comments that I share below. I believe that the proposed method’s interest for practitioners could be substantially increased by demonstrating that the choice of rank K has minimal impact on the obtained estimate (if true), or providing an automatic selection rule for the choice of rank K (perhaps based on the computed error bounds?). I would encourage the authors to be more precise in their characterization of low-rank data. Indeed, although the authors mostly emphasize the rank of X, it is not sufficient for X to be well-approximated by a low-rank matrix for the estimates to be accurate, as regression inherently captures the relationship between X and y. Concretely, consider a ridge regression problem with a dataset with N = 200, D = 100 where each x is drawn from a mixture between a spherical gaussian on the first 10 dimensions (with probability 0.9) and a spherical gaussian on the last 90 dimensions (with probability 0.1). It is not difficult to see that the covariance (and empirical covariance) matrix are well-approximated by the first 10 eigenspaces. Using K = 10 would lead to mostly vacuous bounds on the estimation of Q_n for observations that are from the second mixture component. However, if the response variance is much larger for this mixture component, the observations in the second component will account for a large part of the total m.s.e., rendering the low-rank approximation inaccurate. ============== added after response ================== the authors are correct in pointing out that proposition 3 imply that the procedure minimizes the average errors on the Q_n. Unfortunately, this does not necessarily lead to good estimates for the NS method, as in a general sense, we wish to minimize the average error on 1 / (1 - Q_n), hence why the example I constructed could be problematic. Very concretely, suppose that we are considering least-squares regression, then we have that y - y_NS = (y - \hat{y}) / (1 - Q). We see that for a good approximation, we would thus in fact wish to pick the subspace we discarded, as it contains observations with the highest leverage which is important to approximate. ================================================= It would be interesting to discuss the ALR approximation also in the context of sparse estimators for which ACV methods have also garnered interest (e.g. as discussed in Rad and Maleki, 2020, or Wang et al. 2018), as they are of particular relevance when the data dimension is large and are in common use by practitioners. Indeed, they also often benefit from a (different type of) computational advantage, depending not on the ambient dimension D but the dimension of the estimate which may be much lower.


Review 3

Summary and Contributions: The paper considers the problem of approximating the Cross Validation (ACV) error in Generalized Linear Models (GLMs) with L2-regularization. This is a topic that has been studied by a number of authors in the recent years with [Wilson, Kasy, Mackey'20] providing the most general non-asymptotic bounds to date. The main bottleneck that ACV methods suffer from is the inversion of the Hessian that in general requires $O(d^3)$ operations. Furthermore, the approximation quality of such methods typically degrades as the dimension of the data $d$ increases. To overcome the above limitations the paper proposes a method to bound the approximation error of two popular ACV schemes that exploits (Approximate) Low Rank structure of the data. To achieve this, the authors follow the proof strategy from [Wilson, Kasy, Mackey'20] to get efficiently computable bounds that incorporate the effect of having approximately low rank data. For relatively large values of the regularization parameter $\lambda \geq 1$ and Poisson Regression, the authors provide experimental evidence that their method provides significant improvements in the time required to execute ACV without sacrificing accuracy/error significantly. ---------------------------------------------------------------------------------------------------------------- >> Update: in my opinion the authors have satisfactorily addressed most of the points raised by the reviewrs. Though the paper leaves quite a few things to be desired from a theoretical stand-point, I do believe the approach taken here is original, the data-dependent error bounds reasonably easy to evaluate and that the paper can inform the practice of ACV. Thus, I am happy to change my overall evaluation to Accept. ----------------------------------------------------------------------------------------------------------------

Strengths: [S1] The main strength of the paper is the potential practical significance that the method may have for ML practicioners in accelerating the CV procedure while also providing interpretable and computationally efficient approximation bounds.

Weaknesses: [W1] The main weakness of the paper in my opinion is the limited experimental evaluation of the method in particular with respect to the range of values of the regularization parameter that the ACV is applied for. In practice, CV is applied to a wide-range of regularization parameters in order to arrive at a specific value. For the specific method proposed here this is more significant as the approximation error upper bounds derived here depend inversely on the regularization parameter (Proposition 4). Thus, it is very plausible that the method proposed here is only effective for relatively large values of the regularization parameter $\lambda \geq 1$. ---------------------------------------------------------------------------------------------------------------- >> Update: in their respose, the authors include further experimental evaluation showing that although indeed the performance of the method degrades for smaller lambda the effect is significant when the target rank K is close or smaller than the true approximate low rank R. I thank the authors for going through the effort on doing the experiments and clarifying this point. My conclusion is that their method seems to be reasonably robust for typical values of \lambda. ---------------------------------------------------------------------------------------------------------------- [W2] At a secondary level, a weakness of the paper is that the effect of the data being approximately Low Rank (ALR) versus Exactly Low Rank is not quantified theoretically. This is significant as in practice data are very rarley exactly low rank and the recent work of [Udell, Townsend 2019] that is used as motivation for the paper use a non-standard notion of ALR through the element-wise supremum norm of the matrix. Ideally, a paper of this type should give interpretable bounds that directly express the effect of being approximately low rank vs exactly low rank. ---------------------------------------------------------------------------------------------------------------- >> Update: even after reading the authors' response, I still believe that within their current framework it will be challenging to provide fully formal guarantees for a specifc notion of ALR. Nevertheless, my viewpoint is that the paper is interesting enough and practicaly relevant so this deficiency is of lesser significance. ---------------------------------------------------------------------------------------------------------------- [W3] Finally, the method proposed in the paper to obtain a low rank approximation of the Hessian (Algorithm 2), uses a diagonal approximation of the Hessian to approximate an optimal choice, the effect of which and the failure modes is never discussed and quantified. ---------------------------------------------------------------------------------------------------------------- >> Update: the authors' response addresses this point in a reasonable, though, non-rigorous manner. Given that the main interest of the paper is towards building a practical and scalable ACV method with some data-dependent and computable guarantees, I deem the response satisfactory. ----------------------------------------------------------------------------------------------------------------

Correctness: The claims are for the most part correct. There is a little bit of uncertainty about the validity of Proposition 4, that hopefully the authors can clarify. ---------------------------------------------------------------------------------------------------------------- >> Update: the authors have addressed the question in their response in a satisfactory manner. ----------------------------------------------------------------------------------------------------------------

Clarity: The paper is well written, the problem they consider is well defined and motivated. The description of the experimental evaluation has room for improvement.

Relation to Prior Work: Prior work is adequately discussed and cited.

Reproducibility: Yes

Additional Feedback: - [pg 2, ln 64]: ``First we prove that existing ACV methods automatically obtain high accuracy in the presence of high-dimensional yet ALR data". This statement has not been substantiated by the theory presented in this paper. [ALR] Approximate Low Rank is quantitative statement that requires a precise definition. Typically this is either phrased in terms of $\ell_{p}$ norms of the vector of the $d-k$ smaller singular values of the matrix or more recently [Townsend, Udell] the notion of approximate low rank was in terms of the elementwise $\ell_{\infty}$ norm of the residual matrix. The paper does not quantify the error bounds of ACV in terms of any specific notion of low rank and effectively only argues about the exact low rank case (Corollary 1). - [pg 2, Figure 1]: the sequence of labels in the legend makes it a little bit difficult to read the figure. My suggestion is to make the blue label appear first in the legend. -[pg 3, ln 111]: ``tend to either focus on low dimensions or show poor performance in high dimensions", here this statement is not qualified enough. It would be helpful to give an approximate scaling law for example from [Wilson, Kasy, Mackey'2020] to give more intuition. - [pg 3, ln 91]: definition of $ \hat{D}_n ^ (k)$, it is probably better to define this only for $k=1,2$ as for $k=3$ the definition (ln 132) is slightly different and it causes confusion to the reader. - [pg 6, ln 212]: "...choice of $\Omega$ in Algorithm 2 approximates the optimal choice $H^{-1} V_K$". For diagonal dominant matrices this choice is well motivated, however in general it is not clear how well this approximation behaves. This raises the question that is not answered in the paper, [Q1] What are the failure modes of this approximation/technique (Algorithm 2, ln 5) and what general bounds can be provided? - [pg 7, Proposition 4]: in the proof (Appendix D.1), the following fact is stated without justification " $b_n$ is parallel to $v_D$ it must be that $\gamma_{D} \geq ||b_n||^{2}$". I don't see how this follows from the assumptions. The same assertion is being made in lines 223-224. [Q2] Why is the assertion " $b_n$ is parallel to $v_D$ it must be that $\gamma_{D} \geq ||b_n||^{2}$" true? - [lines 238, 252]: the values of the regularization parameter $\lambda$ chosen are arbitrary and relatively ``large", e.g. $5.0$ and $1.0$. In practice, CV and ACV is applied to a range of values for the parameters in order to perform model selection. Depending on the problem such large values might not be realistic. Coupled with the fact that the bounds in Proposition 4, depend inversely on lambda, this raises the question of the robustness of the method as a function of lambda. [Q3] Is the method robust for a wide range (e.g. $\lambda \in (0.01, 100)$) of values of the regularization parameter that typically practitioners search over? The experiments need to show whether the estimates are accurate enough for smaller values than $1$. - [pg 8, Figure 3]: it is not clear what is being compared here. Is it CV, IJ, $\tilde{IJ}$ applied to the squared-loss for the estimate given by eq (1)? In conclusion, I would be willing to significantly improve my score if the questions raised here are adequately addressed by the authors. With order of importance [Q2] > [Q3] > [Q1].

[Author Response · NeurIPS 2020]

We are grateful to the reviewers for their helpful feedback. Recall that, in this paper, we observe that existing
approximate cross-validation (ACV) methods may be slow and inaccurate in GLM problems with high data dimension
($D$). To address this issue, we provide a new ACV method, which we show is both fast *and* accurate for approximately
low-rank (ALR) data. And we provide an efficiently computable upper bound on the error of our ACV method.

We agree with **R1** that we do not focus on asymptotic analysis. We see our focus on finite-sample bounds as a strength
rather than a deficit. By considering the data at hand rather than an imagined infinite population (which may often
be subject to misspecification concerns), we provide computable bounds that users can apply to their own data, with
accuracy guarantees for their particular application. That being said, we do provide some conditions under which our
bounds asymptotically go to zero (and so must be tight) in Corollaries 1 and 2. We believe a full treatment of bound
asymptotics would be a major undertaking and out of scope of the current paper.

**R1** suggests using principal components analysis (PCA) to reduce dimensionality of the covariate matrix. We see two
potential interpretations of this suggestion. (1) We care about the full high-dimensional GLM, but we could use PCA as
an approximation within existing ACV methods — as an alternative to our approximation. (2) Every problem is already
low-dimensional because practitioners always apply dimensionality reduction (e.g. PCA) first. The issue in both cases
is that PCA is task agnostic; e.g. the covariates associated with a response need not be in the first components of PCA
[Jolliffe 1982]. So, for (2), there is real interest in the full-covariate GLM. For (1), we note that our method does encode
task, namely by reducing the Hessian (a function of both covariates and predictors) via its associated quadratic form
(lines 91–92). We will illustrate with an empirical comparison to the proposal in (1) in our revision.

**R1** suggests citing "Nouridine to illustrate the importance of leave-one-out estimators for prediction risk estimation."
We will provide more discussion of the benefits of LOOCV (vs $K$-fold CV) as shown in e.g. Figure 1 of Rad and Maleki
[2020]. We searched for relevant papers by (e.g.) Noureddine El Karoui but did not find one. If R1 could provide an
exact citation, we would be happy to include any appropriate references.

We agree with **R3** that having a hyperparameter $K$ is undesirable. However, we note that $K$ is different from most
"tuning parameters," where too-large or too-small values are both bad. For $K$ we instead recommend that practitioners
use the largest $K$ allowed by their computational budget. Crucially, our cheaply computable error bounds allow a
practitioner to check if this $K$ allows sufficient accuracy.

We agree with **R3 and R4** that we should be more clear what is meant by ALR data. In all rigorous mathematical
statements, we use the typical definition of ALR corresponding to having only a few large singular values (e.g. Corollary
1). We will add a more exact statement and discussion early in the paper. As **R4** notes, the issue is compounded by our
citation of Udell and Townsend [2019], who use a different ALR definition. Despite the difference, we still believe their
work helps motivate why many matrices are ALR in the spectral sense (cf. the success of our method on real datasets).
We will clarify this point in the paper. **R3** also suggests we may need additional assumptions to estimate $Q_n$ well with
our method. While the example given by R3 shows that using the top eigenvectors and eigenvalues can lead to poor
estimates of some $Q_n$, we note that Prop. 3 implies this issue cannot happen *on average*. As our interest is in computing
$Q_n$ for all $n$, we believe that Prop. 3 does show that $H$ being ALR is sufficient for the success of our method.

36–47 **R4** asks [Q2] about Prop. 4 in App. E.1. Recall $\gamma_d$ are the eigenvalues of $\sum_n b_n b_n^T$; so $\gamma_D = \sum_n \langle b_n, v_D \rangle^2$. If $b_n \propto v_D$ for some $n$, we have $\gamma_D \geq \langle b_n, v_D \rangle^2 = \|b_n\|_2^2$. We will clarify this logic.

**R4** observes [Q3] that our experiments use $\lambda \geq 1$ and wonders about $\lambda < 1$. We chose $\lambda$ to make the optimization problems sufficiently regular, so exact CV runtime would be reasonable for the larger experiments. Still, we appreciate R4's caution. To address dependence on $\lambda$, we ran a $N = 600, D = 400$ synthetic logistic regression task of approximate rank $R = 200$ (the singular values follow $\sigma_{200} \approx 10.0, \sigma_{201} \approx 1.0$) for a range of $\lambda$ values. In the figure, we show percent error in the estimate of exact CV, $x_n^T \hat{\theta}_{\backslash n}$, via the NS approximation across different settings of $K$ and $\lambda$. The errors'
48 change with $\lambda$ is minimal, except for values of $K$ near $R$. We will include real-data experiments for the "low-$\lambda$" regime
49 in our revision.

50 **R4** asks [Q1]: what are the failure modes of the approximation in Algorithm 2, line 5? This approximation has two
51 parts: (1) using a single iteration of the subspace iteration method and (2) a diagonal approximation to the Hessian.
52 For (1), the subspace iteration method converges in the fewest iterations when there is a sharp dropoff after the first $K$
53 singular values. For (2), the Hessian is the least diagonal when the covariates are all linearly dependent and completely
54 diagonal when they are orthogonal. We will expand on these points with experiments in an updated version of the paper.
55 ■ *References*: I. T. Jolliffe. A note on the use of principal components in regression. Applied Statistics. 1982.

[Meta-Review · NeurIPS 2020]

Two reviewers agree that this submission represents an important contribution to the field. However, a third expressed significant concerns about the tightness of the presented bounds, the accommodation of matrices with growing rank, and behavior in the presence of principal component preprocessing. Please be sure to carefully review and address the concerns of all reviewers in the revision.